# Probing altered receptor specificities of antigenically drifting human H3N2 viruses by chemoenzymatic synthesis, NMR, and modeling

Luca Unione [1,2,3] ✉, Augustinus N. A. Ammerlaan[1], Gerlof P. Bosman[1], Elif Uslu [1], Ruonan Liang[1], Frederik Broszeit[1], Roosmarijn van der Woude[1], Yanyan Liu[1], Shengzhou Ma[4], Lin Liu[4], Marcos Gómez-Redondo[2], Iris A. Bermejo[2], Pablo Valverde[2], Tammo Diercks[2], Ana Ardá [2,3], Robert P. de Vries [1] ✉ & Geert-Jan Boons [1,4,5,6] ✉

Prototypic receptors for human influenza viruses are *N*-glycans carrying α2,6-linked sialosides. Due to immune pressure, A/H3N2 influenza viruses have emerged with altered receptor specificities that bind α2,6-linked sialosides presented on extended *N*-acetyl-lactosamine (LacNAc) chains. Here, binding modes of such drifted hemagglutinin's (HAs) are examined by chemoenzymatic synthesis of *N*-glycans having ¹³C-labeled monosaccharides at strategic positions. The labeled glycans are employed in 2D STD-¹H by ¹³C-HSQC NMR experiments to pinpoint which monosaccharides of the extended LacNAc chain engage with evolutionarily distinct HAs. The NMR data in combination with computation and mutagenesis demonstrate that mutations distal to the receptor binding domain of recent HAs create an extended binding site that accommodates with the extended LacNAc chain. A fluorine containing sialoside is used as NMR probe to derive relative binding affinities and confirms the contribution of the extended LacNAc chain for binding.

Spill-over of influenza A virus (IAV) from an animal reservoir into the human population has caused four pandemics in the past century[1]. These viruses became seasonal strains that evade neutralization induced by prior infections or immunization. Seasonal IAV causes significant disease burden, infecting 9–35 million individuals annually and causing 12–56 thousand deaths[2].

IAV expresses hemagglutinin (HA) and neuraminidase (NA) as two major envelope glycoproteins. HA binds to sialic acid (Neu5Ac) of glycoconjugates on host cells to initiate infection, whereas NA facilitates the release of progeny viruses from infected cells by cleaving sialosides[3]. Human IAVs recognize sialosides α2,6-linked to galactoside (Gal), which are structures that are abundantly found in the upper respiratory tract of humans. On the other hand, HAs of ancestorial avian viruses exhibit a preference for α2,3-linked sialosides that are found in the duck enteric and chicken upper respiratory tract[4–8].

Human IAVs have a remarkable ability to evolve and evade neutralization by antibodies elicited by prior infections or vaccination. This antigenic drift is mainly caused by mutations in the receptor

[1]Department of Chemical Biology & Drug Discovery, Utrecht Institute for Pharmaceutical Sciences, Utrecht University, 3584 CG Utrecht, The Netherlands. [2]CICbioGUNE, Basque Research & Technology Alliance (BRTA), Bizkaia Technology Park, Building 800, 48160 Derio, Bizkaia, Spain. [3]Ikerbasque, Basque Foundation for Science, Euskadi Plaza 5, 48009 Bilbao, Bizkaia, Spain. [4]Complex Carbohydrate Research Center, University of Georgia, 315 Riverbend Rd, Athens, GA 30602, USA. [5]Bijvoet Center for Biomolecular Research, Utrecht University, Utrecht, The Netherlands. [6]Department of Chemistry, University of Georgia, Athens, GA 30602, USA. ✉e-mail: lunione@cicbiogune.es; R.Vries@uu.nl; g.j.p.h.boons@uu.nl; gjboons@ccrc.uga.edu

binding site at the globular head of HA[8–10]. A/H3N2 viruses, which are the leading cause of severe seasonal influenza illness[11] resulted in altered glycan binding properties. These changes were first noticed by a lack of agglutination of fowl red blood cells. Furthermore, these viruses also propagate poorly in common laboratory hosts such as MDCK cells and eggs, indicating they require glycan receptors that are not expressed by these cells or hosts[12–14]. Glycan microarray and other binding studies have indicated that these viruses have lost the ability to bind to simple α2,6-linked sialoglycans[15] and instead bind to bi-antennary N-glycans that have on both arms extended N-acetyl lacto-samine (LacNAc) moieties terminating in an α2,6-linked sialoside. Computational studies indicated a binding mode in which the two sialosides of the bi-antennary N-glycan bridge the binding sites of a trimeric HA protein resulting in high avidity of binding[15].

We have constructed a glycan microarray populated with bi-antennary N-glycans that more closely resemble structures expressed by human respiratory tissue[16]. It includes N-glycans having multiple consecutive LacNAc repeating units that can be capped by sialic acid. Glycomic analysis has shown that these structures are abundantly expressed by a human epithelial cell line[16] and respiratory tissues[17,18] and are expected to be relevant receptors for IAVs. Compound 1 (Fig. 1) was established as the minimal epitope for early A/H3N2 viruses, whereas those that emerged after 2000 (e.g., NL2003) did not bind to 1 and required as minimal epitope a compound such as 2 with one of its arms presenting a tri-LacNAc moiety having a terminal α2,6-linked sialoside[19]. Microarray data combined with sequence alignment, reported crystal structures, and MD simulations indicated an alter-native binding model for recent A/H3N2 of the 3 C.2 clade in which mutations remote from the receptor binding domain resulted in a rotation of the side chain of Tyr159, creating an extended binding site[16]. Furthermore, a 225 G/D mutation reoriented the poly-LacNAc chain of the compound allowing it to establish interactions with this extended binding site.

In this work, we describe the chemoenzymatic synthesis of N-glycan 2 and the [13]C-labeled analogs 3, 4, and 5 that differ in the positions of [13]C-labeled monosaccharides along the poly-LacNAc chain. The [13]C-labeled glycans are employed in 2D saturation transfer difference (STD)-[1]H,[13]C-HSQC experiments to pinpoint those mono-saccharides of the poly-LacNAc chain of 2 that engage with HAs from evolutionarily distinct A/H3N2. The NMR data in combination with

in silico and mutagenesis studies demonstrate that mutations distal to the receptor binding domain of HA create an extended binding site that can directly interact with the extended LacNAc chain of glycans. To investigate whether the affinity of HAs of contemporary A/H3N2 strains for the extended glycans has improved or compensated due to mutations at the receptor binding site, we employ a fluorine-containing sialyl-LacNAc derivative ([19]F-SLN, compound 6 in Fig. 1) as a [19]F NMR probe to derive relative binding affinities. This demonstrates that the additional contributions of the LacNAc chain improve binding affinity only for recent A/H3N2 strains.

The preparation of isotopomers 3–5 is critical for the STD NMR experiments and uncover the contributions of each monosaccharide of the tri-LacNAc chain of 2 for binding to various HAs. This type of NMR experiment is a powerful approach to characterize glycans interactions with proteins at the atomic level[20]. Ligand protons in close proximity to the protein receive saturation resulting in strong STD signals. On the other hand, ligand protons that are more remote or do not engage with the protein exhibit weak or no STD signals. Large glycans suffer, however, from very low chemical shift dispersion of their [1]H NMR signals precluding detailed binding studies by [1]H-STD NMR. This is especially problematic when glycans present repeating units, such as poly-LacNAc chains, as these have almost indistinguishable [1]H chemical shifts. These difficulties can be alleviated by using [13]C isotope labels introduced at specific positions of an oli-gosaccharide chain. [13]C-filtered NMR spectra are then specific for these site-selective probes and free of background signals from unlabeled monosaccharides within the N-glycan or from the HA glycoprotein. Such isotope labeling preserves binding characteristics while allowing heteronuclear NMR studies with atomic resolution and substantially increased sensitivity[21,22]. A challenge that we address in this study is the preparation of high-complex glycan having selectively [13]C labeled monosaccharides.

## Results

### Synthesis of asymmetrical [13]C labeled N-glycans

Glycan 2 and the three asymmetric isotopomers 3-5 (Fig. 1) were pre-pared from glycosyl asparagine 8 that could readily be prepared from a sialoglycopeptide extracted from egg-yolk powder followed by pro-nase treatment and then acid-mediated hydrolysis of the sialosides (Fig. 2a)[23]. UDP-GlcNAc and UDP-Gal having [13]C-isotopes at carbons 1 to 6 in combination with appropriate glycosyl transferases were employed to introduce [13]C-labeled GlcNAc and Gal moieties into the target oligosaccharides. To selectively extend the α(1,3)-Man arm of 8 by an oligo-LacNAc chain, we exploited the inherent branch selectivity of β-galactoside α2,6-sialyltransferase 1 (ST6Gal1), which has a 10-fold higher activity for the α3-arm[24], allowing for the convenient prepara-tion of 9. Next, the remaining terminal galactoside at the α6-arm of 9 was temporarily modified by an α1,2-fucoside by treatment with α1,2-fucosyltransferase 1 (FuT1) in the presence of GDP-Fuc to give 10. The α1,2-fucoside temporarily blocks the α6-arm from modifications by mammalian glycosyl transferases[25], and thus after removal of the α2,6-sialoside of 10 by acid-mediated hydrolysis, the α3-arm of the resulting 11 could be selectively elaborated into a tri-LacNAc chain by the sequential use of B3GnT2 and B4GalT1 in combination with UDP-GlcNAc and UDP-Gal, respectively. The use of the [13]C-labeled sugar donors for the installation of the first or second LacNAc moiety gave access to compounds 15 and 20 having the terminal and central Lac-NAc moiety labeled by [13]C-isotopes, respectively (Fig. 2b, c). Target glycans 3 and 4 were obtained by treatment 15 and 20 with ST6Gal1 in the presence of CMP-Neu5Ac to give 16 and 21 that were treated with an α1,2-fucosidase to provide target compounds 3 and 4.

To introduce the [13]C labeled galactoside in the inner LacNAc moiety, symmetrical N-glycan 8 was treated with galactosyl hydro-lase from E. coli that has high preference for the α3-arm[26], to give after purification by HILIC-HPLC, asymmetrical glycan 22 (Fig. 3). The

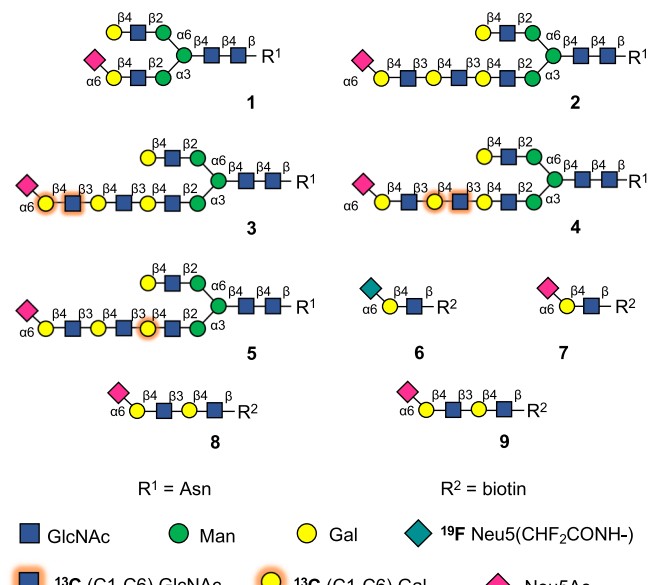

**Fig. 1 | Synthesized sialoglycans.** Isotopically glycans for NMR studies with evo-lutionary distinct HAs.

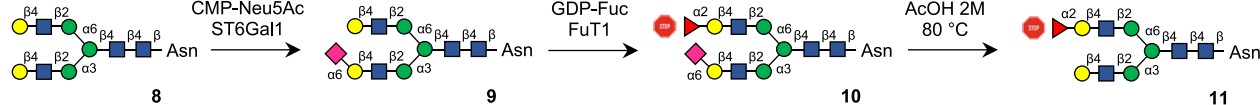

**a. Preparation of common precursor 11**

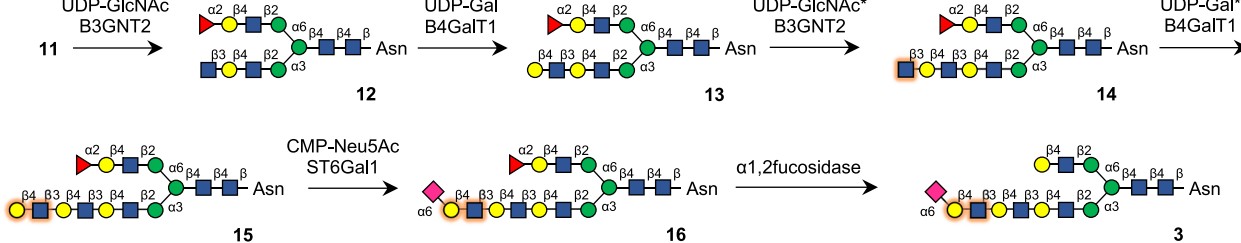

**b. Preparation of asymmetrical glycan 3 having terminal ¹³C LacNAc moiety**

**c. Preparation of asymmetrical glycan 4 having a central ¹³C LacNAc moiety**

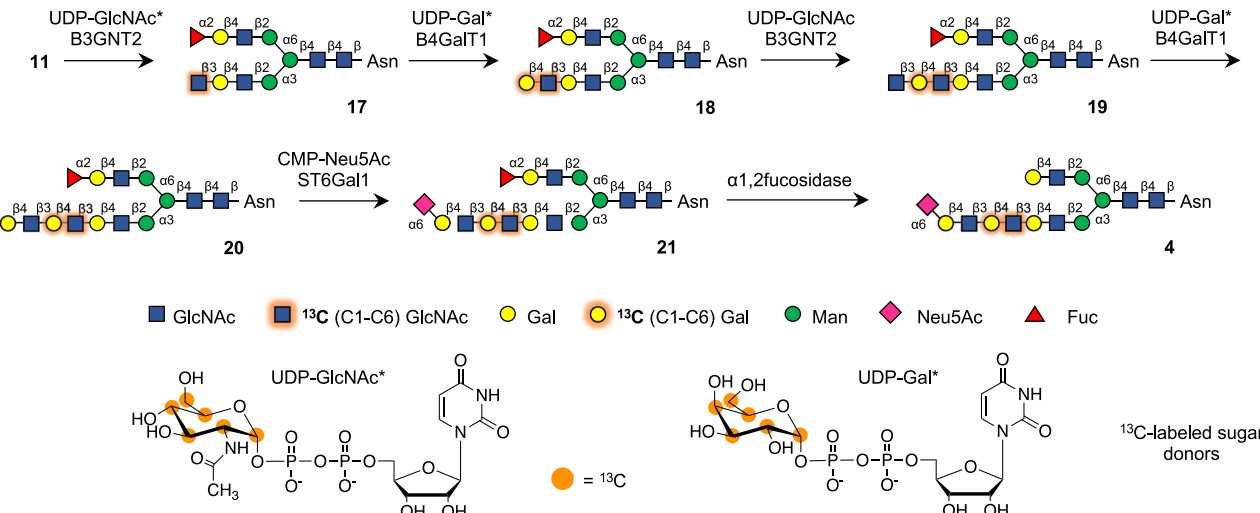

**Fig. 2 | Chemoenzymatic synthesis of ¹³C labeled sialoglycans 3 and 4. a** The α(1,3)-arm of glycan **8**, which was selectively modified as α2,6-linked sialoside allowing selectively fucosylation of the α(1,6)-arm which temporarily block it glycosyl accepting properties (stop). After acid hydrolysis of the 2,6-linked sialoside, compound **11** was obtained that allows selective extension of the α(1,3)-arm.

**b** Glycan **3** was prepared by employing ¹³C labeled UDP-GlcNAc and UDP-Gal during the introduction of the final LacNAc assembly followed by treatment sialylation with ST6Gal1 and removal of the fucoside using a 1,2-fucosidase. **c** Glycan **4** was prepared by using ¹³C labeled UDP-GlcNAc and UDP-Gal during the introduction of the central LacNAc moiety.

terminal galactoside at the α6-arm of **22** was temporarily blocked as α2,3-linked sialoside by treatment with β-galactoside α2,3-sialyl-transferase 4 (ST3Gal4) in the presence of CMP-Neu5Ac to provide **23**. Next, a ¹³C-labeled galactoside was linked to the terminal GlcNAc moiety of **23** by employing B4GalT1 and ¹³C-labeled UDP-Gal to give **24**. The α3-arm of the latter compound was enzymatically extended into a tri-LacNAc structure to give **28** by repeated use of B3GnT2 and B4Gal1. Treatment of **28** with ST6Gal1 in the presence of CMP-Neu5Ac installed an α2,6-sialoside thereby providing **29**. Finally, the α2,3-linked sialoside at the α6-arm of **29** was selectively removed by treatment with α2,3-neuraminidase to afford target glycan **5**. The same synthetic strategy was used to synthesize unlabelled **2** from unlabeled UDP-sugar donor.

## 1D ¹H-STD NMR experiments

HAs from evolutionarily different time points, HK68 (1968), NL91 (1991), and NL03 (2003), were selected to analyze their binding preferences for unlabeled glycan **2** by 1D ¹H-STD NMR experiments (Fig. 4a). HK68 is the prototypical HA of the 1968 Hong-Kong

pandemic that was adapted from its avian predecessor to bind human-type receptors[27]. NL91 HA was isolated when these viruses could still hemagglutinate fowl erythrocytes. Later in the antigenic evolution, these viruses lost this property as was observed for in the NL03 strain.

First, we examined the receptor binding specificities of HK68, NL91, and NL03 by glycan microarray technology. Thus, a series of *N*-glycans was printed on NHS-activated microarray slides which was probed by several plant lectins for quality control (Supplementary Fig. 4). Next, whole viruses of HK68, NL91, and NL03 were exposed to the glycan microarray, and binding was determined by using broadly neutralizing antibody CR8020 and an Alexa labeled goat-anti-human antibody (Supplementary Fig. 4). HK68 displays some residual avian-type receptor binding specificity and bound 2,3- as well as 2,6-sialo-sides. NL91 only bound human-type receptors including compound **1** and **2** that display the 2,6-linked sialoside at a mono- and tri-LacNAc moiety, respectively. The further evolved NL03 only bound compound **2**. This virus prefers the sialoside at the extended LacNAc moiety to be displayed at the α3-mannose antenna as the isomeric compound did

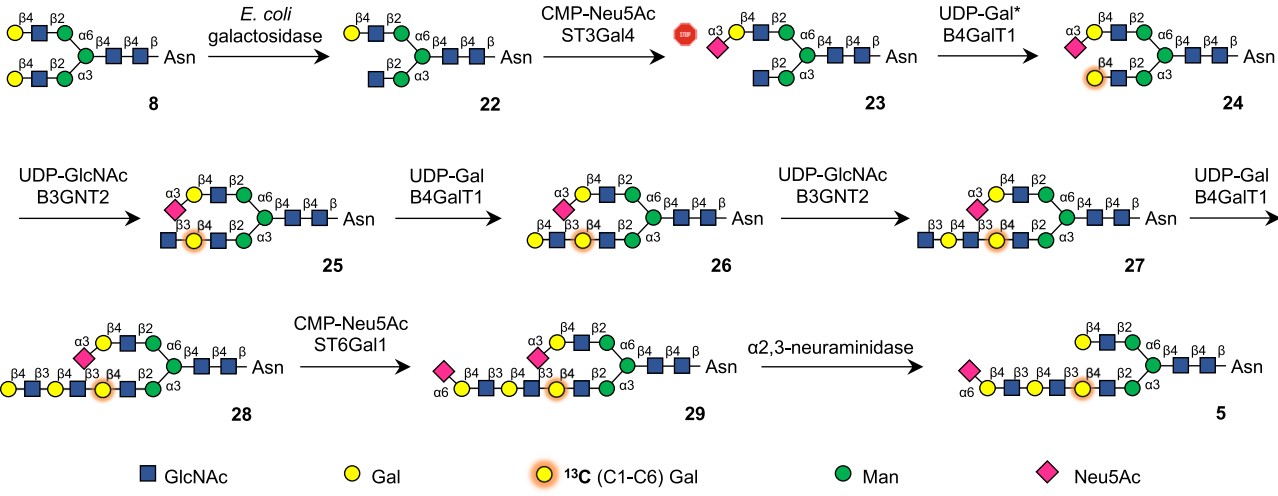

**Fig. 3 | Chemoenzymatic synthesis of glycan 5.** Selective removal of the galactoside of the α3-arm allowed the selective installation of the $^{13}$C-labeled galactoside at the first residue of the tri-LacNAc moiety to give entry into the synthesis of glycan **5**.

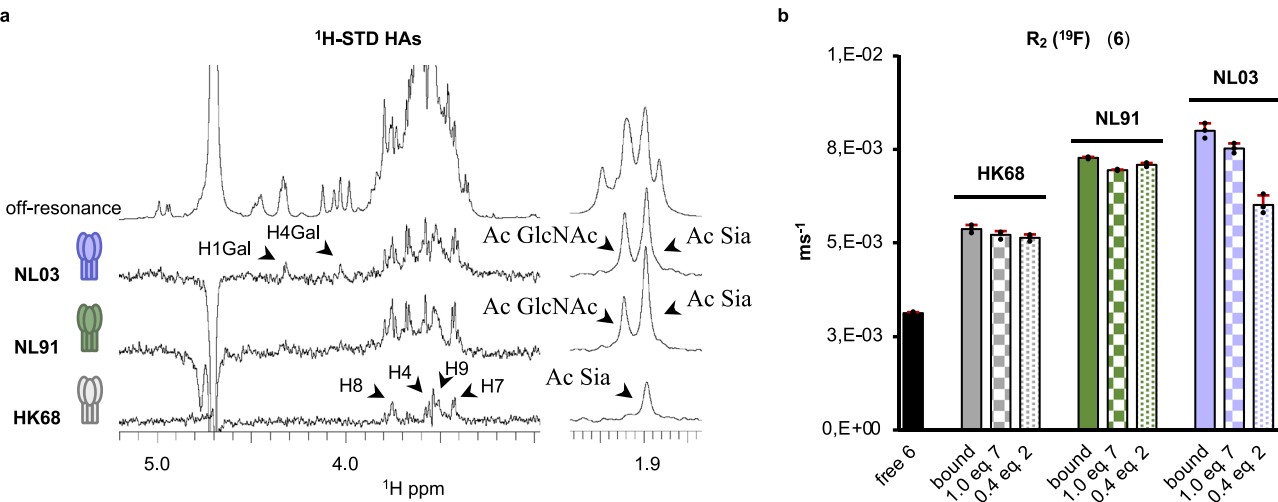

**Fig. 4 | NMR binding studies. a** 1D $^1$H-STD NMR. Spectra obtained for the complexes of HK68, NL91, and NL03 HAs with the glycan **2**, and inset of the acetyl region. **b** $^{19}$F-R$_2$ NMR. Comparison of transverse $^{19}$F relaxation rate, R$_2$($^{19}$F), for the fluorinated probe (**6**) in the absence or presence of the different HA proteins.

Addition of competitor molecules **7** and **2** causes a decrease in R$_2$($^{19}$F) proportional to their affinities relative to the probe compound **6**. The bars represent the average of three replicates ± SD.

not show binding. Mono- and di-sialylated structures gave similar responsiveness indicating that bidentate binding is not important for recognition. Control compounds lacking sialic acid did not bind.

Next, STD experiments were performed using compound **2** and recombinant HAs from HK68, NL91, and NL03. The STD signals confirmed that all three HAs have functional binding properties. As expected, severe $^1$H chemical shift overlap limited $^1$H-STD-based analysis that nonetheless revealed clear differences. Thus, HA from HK68 showed STD signals only for the sialic acid moiety, indicating that only this residue contributes to protein binding. The strongest STD signals were observed for H4, H7, H8, H9, and the methyl group of the acetamide moiety of the sialoside. In contrast, the $^1$H-STD spectrum of NL91 HA showed several additional STD signals, including the methyl group of an acetamide of a GlcNAc residue. Finally, the $^1$H-STD spectrum of NL03 HA was substantially more complex and showed, among others, STD signals above 4 ppm that correspond to the H1 and H4 of galactosides. It was, however, not possible to identify specific monosaccharides due to the chemical shift degeneration of the nuclei of the tri-LacNAc chain. These results indicate that from 1968 to 2003, A/H3N2 HAs evolved to recognize

additional residues of an extended LacNAc chain beyond the terminal α2,6-sialoside.

## Transverse relaxation rate R$_2$($^{19}$F) NMR experiments to establish relative binding affinities

To identify the minimal binding epitope and derive relative binding affinities for the three HAs, we used fluorine-containing Neu5DFA(α2,6)LacNAc ($^{19}$F-SLN, compound **6**, Fig. 1) as probe (see supplementary section 2 for details of the synthesis of **6**). The strategic incorporation of a non-endogenous fluorine atom ($^{19}$F) enables ligand-protein interaction studies by molecule-specific $^{19}$F NMR experiments monitoring R$_2$($^{19}$F)-relaxation rates that do not suffer from overlap or dispersion problems as in $^1$H-based NMR experiments described above[28,29]. First, the overall transverse relaxation rate was measured, R$_2$($^{19}$F), of $^{19}$F-SLN in the absence of HA. Addition of the HAs to this fluorinated probe caused a large increase in R$_2$($^{19}$F) relative to the free form indicating $^{19}$F-SLN binding (Fig. 4b, filled bars). R$_2$($^{19}$F) increases for NL91 and NL03 were substantially larger than for HK68. These observations indicate a substantial affinity increase for the $^{19}$F-SLN trisaccharide from HK68 to NL91 and slightly increased from NL91 to

NL03 as previously also observed by ELISA assays[16]. Together with the results from the 1D ${}^1$H-STD NMR experiments, it is concluded that the higher affinity of NL91 and NL03 HAs for this trisaccharide is due to additional recognition of the Gal and GlcNAc residues which does not occur for HK68.

As the replacement of two hydrogen atoms by fluorine at the acetamide moiety may alter ligand binding properties[30], the suitability of compound **6** as a probe was examined by a competition experiment using unlabeled sialyl LacNAc **7** (Fig. 1). Thus, 1.0 equivalent of **7** was added to the NMR tube containing the three different HA proteins and ${}^{19}$F-SLN (**6**). Addition of the non-fluorinated competitor caused a small, but significant decay of $R_2({}^{19}$F) (*ca.* 5%) that is consistent with similar and low affinities of compounds **6** and **7** for the three HAs (Fig. 4b, square-pattern filled bars) and confirms they bind at the same site of the HAs. The $R_2({}^{19}$F) decay was simulated which confirmed the observations (Supplementary Section 4.3).

Next, we examined the relative HA affinities for compound **2** that present an α2,6-linked sialoside on its extended tri-LacNAc chain. Thus, 0.4 equivalent of **2** was added relative to the fluorinated probe to samples containing the three different HA proteins and ${}^{19}$F-SLN (**6**), (Fig. 4b, dotted-pattern filled bars). For the HAs from HK68 and NL91, the addition of the competing glycan **2** caused a small reduction of $R_2({}^{19}$F) similar to that induced by trisaccharide **7**, indicating that the two additional LacNAc moieties in **2** do not enhance the binding affinity.

On the other hand, addition of compound **2** to the HA of NL03 and ${}^{19}$F-SLN (**6**) resulted in a substantially larger reduction of $R_2({}^{19}$F) with a decay of *ca.* 30%. This value is consistent with a $K_D$ that is at least an order of magnitude smaller than that of **6** (Supplementary Fig. 5a, b). These results indicate that the inner di-LacNAc moiety of **2** substantially contributes to recognition by NL03 by providing additional intermolecular interactions in agreement with the ${}^1$H-STD results.

${}^1$H-STD competition experiments were performed to quantify the contribution of the extended LacNAc moiety to the $K_D$ of NL03. Thus, the intensity of unique ${}^1$H-STD signal of the di-fluoro acetamide moiety of **6** was monitored as function of different concentrations of competitors **7**, **8**, and **9** (Fig. 1). Consistent with the ${}^{19}$F-NMR experiments, the mono- (**7**) and the di-LacNAc (**8**) containing sialosides bound to the protein with a $K_D$ of ∼600 μM, which is similar to that derived for the fluorinated compound **6**. The $K_D$ of the tri-LacNAc containing sialoside **9** was determined to be 30 μM. These results further support that the underlying tri-LacNAc chain of **9** substantially contributes to binding of NL03, resulting in a 10-fold higher affinity compared to the mono- and di-LacNAc containing sialosides (The corresponding ${}^1$H-STD spectra are reported in Supplementary Fig. 5c).

## 2D ${}^1$H-STD-${}^1$H,${}^{13}$C-HSQC experiments

To elucidate specific contributions of each monosaccharide of the poly-LacNAc chain of **2** to HA binding, we recorded 2D ${}^1$H-STD-${}^1$H,${}^{13}$C-HSQC NMR experiments for isotopomers **3**, **4**, and **5** in the presence of the three HAs. This variant of the ${}^1$H-STD experiment covers ${}^1$H[${}^{13}$C]-STD effects only for protons bound to a ${}^{13}$C isotope and, therefore unambiguously demonstrates whether specific ${}^{13}$C labeled monosaccharides contribute to recognition or not. ${}^1$H-STD-${}^1$H,${}^{13}$C-HSQC experiments with isotopomer **3** showed clear differences in recognition between HK68, NL91, and NL03 HA (Fig. 5a). For HK68, no ${}^1$H[${}^{13}$C]-STD signals were observed indicating that the terminal LacNAc unit is not part of the recognized epitope. In contrast, ${}^1$H[${}^{13}$C]-STD signals were observed with moderate and strong intensities for NL91 HA and NL03 HA, respectively, indicating that these HAs evolved to recognize sialic acid linked to a LacNAc moiety.

Similar experiments with isotopomer **4**, in which the central LacNAc unit is ${}^{13}$C-labeled uncovered further differences in recognition by the three HAs (Fig. 5b and Supplementary Fig. 7). As expected, HK68

HA did not produce any ${}^1$H[${}^{13}$C]-STD signal while the HA from NL91 and NL03 clearly invoked signals. The ${}^1$H[${}^{13}$C] STD signals are derived from both Gal and GlcNAc and were stronger for NL03 HA, where the sum of all STD signal was almost twice as those observed for NL91 HA, indicating that the central LacNAc moiety is more intimately bound by NL03. Finally, ${}^1$H-STD-${}^1$H,${}^{13}$C-HSQC experiments with isotopomer **5** (Fig. 5c and Supplementary Fig. 8), which has a ${}^{13}$C-labeled Gal farthest away from the sialoside, did not show any ${}^1$H[${}^{13}$C]-STD signals for HK68 HA. Thus, this most ancient viral HA recognizes the glycan exclusively through the terminal α2,6-linked sialoside. NL91 HA produced weak ${}^1$H[${}^{13}$C]-STD signals while NL03 HA gave rise to the strongest STD signals, where the sum of ${}^1$H[${}^{13}$C]-STD intensities was, again, almost twice as those observed for NL91. The most intense ${}^1$H[${}^{13}$C]-STD signals from **5** in complex with NL03 HA corresponded to the H3, H4 and H5 protons of the ${}^{13}$C labeled Gal residue (Supplementary Fig. 9). In galacto-configured monosaccharides, these protons point in the same direction, forming a so-called alpha-face of the pyranoside rings, and often participate in CH-π interactions with aromatic amino acids receptor proteins[31,32].

The 2D ${}^1$H-STD-${}^1$H,${}^{13}$C-HSQC experiment in combination with residue-specific ${}^{13}$C labeling of glycan **2** made it possible to disentangle the contributions of individual monosaccharides of the extended poly-LacNAc chain for recognition by three different HAs. Increasing ${}^1$H[${}^{13}$C]-STD signal intensities were observed during evolution of HA indicating more intimate contacts with the poly-LacNAc chain. Thus, for the original pandemic A/H3N2 human virus (HK68) the poly-LacNAc moiety does not contribute to binding, whereas, for later strains, site-specific mutations close to its glycan binding site enable further interactions with the extended glycan chain. For the evolutionarily early NL91 HA, these interactions are mostly limited to the sialic acid-linked galactoside while the more contemporary NL03 HA recognizes almost the entire poly-LacNAc chain.

## Modeling of glycan-HA complexes

To uncover the binding modes of the three HAs, we combined the results from the NMR experiments with in silico modeling (Fig. 6). Full atom 1 μs molecular dynamic (MD) simulations in explicit water were performed for asymmetric *N*-glycan **2** in complex with the three HAs. In agreement with previous X-ray studies[33], the complex of **2** with HK68 HA showed that the recognition of sialic acid is mediated by hydrophobic interactions with amino acids Y98, H183, W153, and L226, as well as by hydrogen bonding interactions with G135-N137, S228, and E190 (Supplementary Fig. 10a). Analysis of the MD trajectory showed that E190 engages the Sia-1 OH9 through hydrogen bonding with an average distance of ∼2.9 Å, while L226 makes hydrophobic interactions with the sialic acid-galactose glycosidic linkage. Beyond the sialic acid residue, the MD simulation did not reveal stable intermolecular interactions and throughout the MD trajectory, the poly-LacNAc chain was flexible and mainly solvent-exposed (Supplementary Fig. 11).

The MD simulation for NL91 HA (Supplementary Fig. 10b and 12) revealed that the S145K mutation at the sialic acid binding site enables a new hydrogen bond interaction with the OH4 of Neu5Ac. To probe the importance of this interaction, we created reverse mutations in NL03. Indeed, the K145 is essential for binding and when mutated to serine almost no response was detected when examined for binding on the glycan microarray (Supplementary Fig. 14). On the other hand, the Y137 side chain in NL91 HA is properly oriented to face the sialic acid-galactose glycosidic bond, which is consistent with the STD signals detected for H4 to H6$_S$ of this Gal residue (Fig. 5a) that were not observed for HK68 HA. We also confirmed the effect of this mutation by creating NL03 N137Y which did not significantly alter glycan binding on the array.

Another relevant NL91 HA mutation is Q189R in the 190-helix where the more extended side chain of the R residue is oriented towards solvent. This leaves space for the acetamide group of the

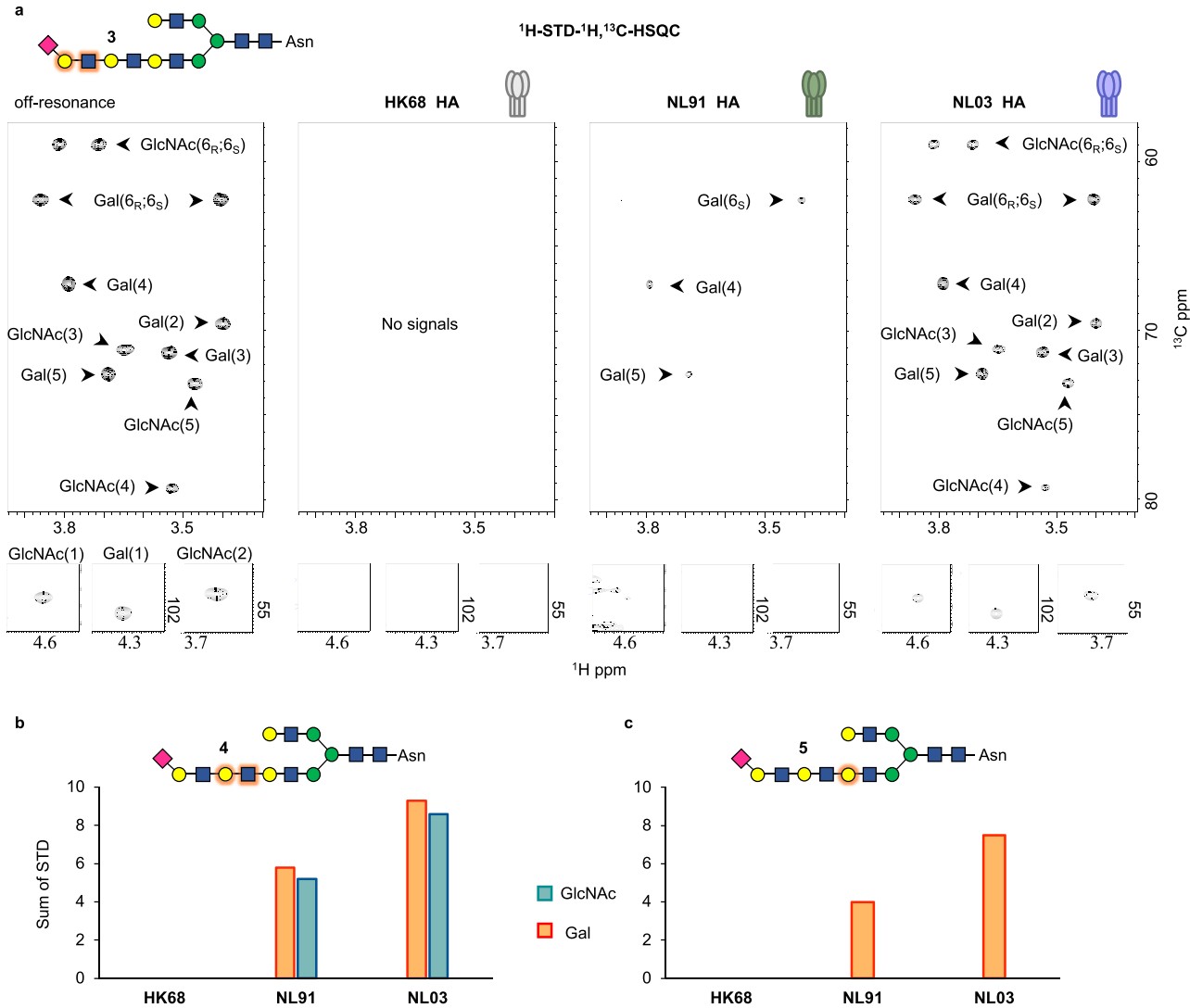

**Fig. 5 | 2D $^1$H-STD-$^1$H,$^{13}$C-HSQC NMR experiments probing the interaction of compounds 3 to 5 with different HA proteins. a** Off-resonance reference (left) and associated $^1$H-STD-$^1$H,$^{13}$C-HSQC spectra for the indicated three HA proteins in complex with compound **3. b, c** Sum of all $^1$H[$^{13}$C]-STD signals of the $^{13}$C labeled Gal (orange) and GlcNAc (cyan) moieties in compounds **4** (b) and **5** (c) in complex with the three HA proteins.

GlcNAc (GlcNAc-3) to be accommodated into a hydrophobic pocket of the protein, where its methyl group makes van der Waals contacts with L194. These results are consistent with the observed $^1$H-STD signal for the acetamide moiety of the GlcNAc residue in NL91 HA (Fig. 4a).

The MD trajectory for NL03 HA in complex with **2** (Supplementary Figs. 10c and 13) showed that the further Y137S mutation restores a hydrogen bond with the carboxylic acid of Neu5Ac as originally observed for HK68[33], while maintaining the K145 hydrogen bond to OH4 of the sialoside, as in NL91. Yet, the concomitant E190D mutation now precludes hydrogen bonding with OH9 of the sialoside due to the shorter D-side chain. The G225D mutation appears crucial to switch the glycan orientation in the receptor binding site consistent with the observed stronger $^1$H-STD signal for the terminal Gal in NL03 relative to NL91 HA and with previous X-ray studies[34,35]. The new hydrogen bonds between D225 side chain and both OH4 and OH3 in Gal forces the α2,6-sialyl-Gal glycosidic linkage in a conformation that moves the underlying glycan chain towards the HA 190-helix (Fig. 6d). In this orientation, the hydrophobic interaction between GlcNAc-3 acetamide moiety and L194 is stabilized because S193 can establish an additional hydrogen bond with Gal-4. Further mutations distal to the receptor binding site (RBS) in NL03 HA reorient the Y159 side chain to face the

Gal-6 and establish CH-π interactions (Supplementary Fig. 10c). The MD results are in accordance with the NMR data showing a substantial increase in $^1$H-STD signal intensities for H3, H4, and H5 of the Gal-6 residue. Thus, four residues (D225, L194, S193, and Y159) in the binding pocket of NL03 HA accommodate the extended glycan chain and contribute to the strong $^1$H[$^{13}$C]-STD signals observed for compounds **3**, **4**, and **5** in the presence of NL03 HA.

## Structural studies using additional variants and mutagenesis to confirm the binding models

We expanded our study by including two additional HAs having further mutations in the extended binding site. NL09 HA (2009) has K145N, Y159F, S193F, and D225N mutations that according to our models are expected to impair recognition of glycan **2**. Indeed, the 2D $^1$H-STD-$^1$H,$^{13}$C-HSQC experiment with the three isotopomers **3**, **4**, and **5** showed substantially weaker $^1$H[$^{13}$C]-STD signals than observed for NL03 HA (Fig. 7b). MD simulation revealed that N145 cannot engage the sialic acid by hydrogen bonding interaction in the way K145 does for NL03 HA (Supplementary Fig. 10c). Furthermore, the substitution of D by N at residue 225 results in a weaker hydrogen bond acceptor while the S193F mutation precludes hydrogen bond interaction with

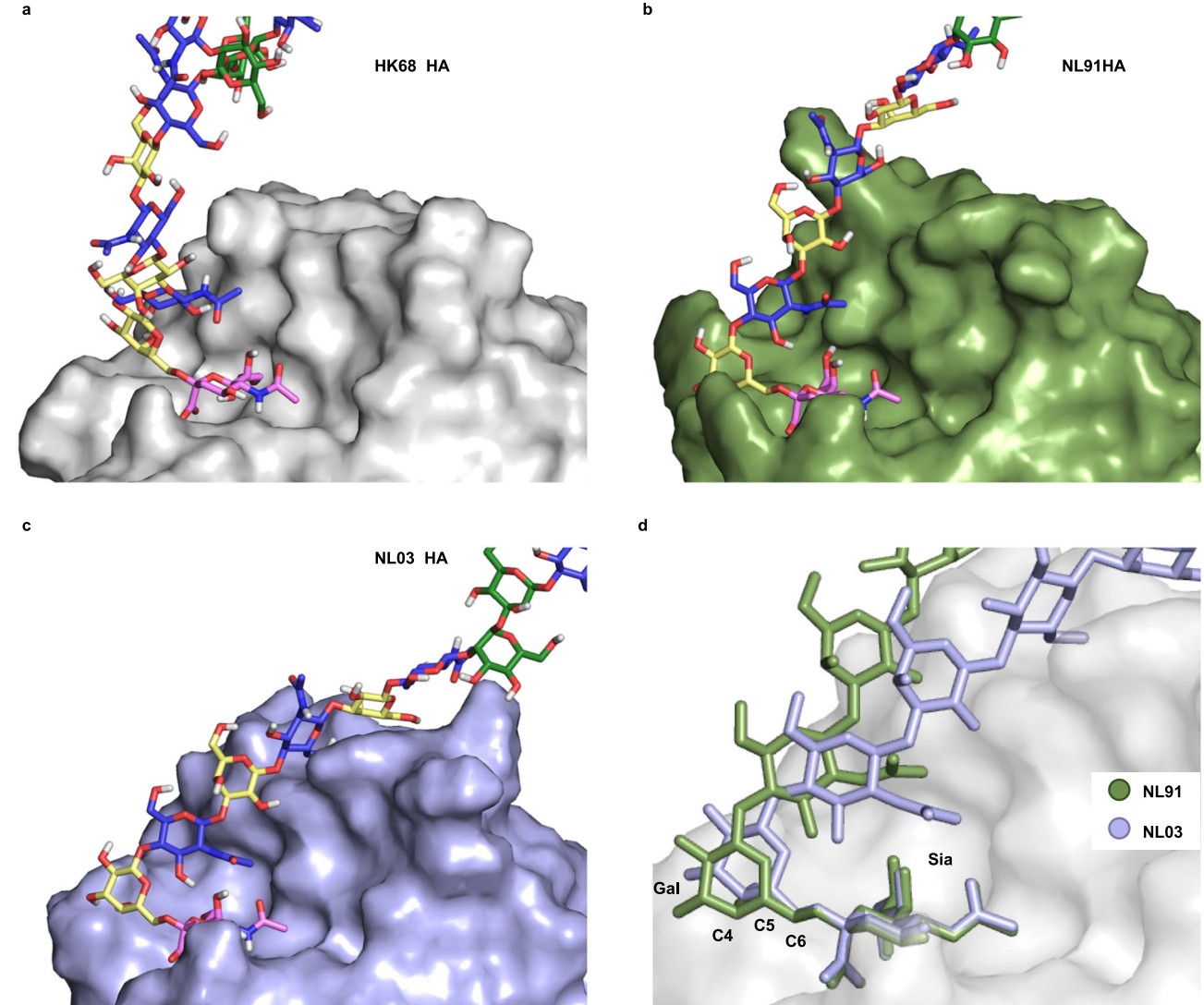

**Fig. 6 | Models of *N*-glycan 2 binding by different HA proteins. a** Binding of **2** by HK68 HA. Only the terminal sialic acid fits into the protein binding pocket while the rest of the poly-LacNAc chain is solvent-exposed. **b** Binding of **2** by NL91 HA. The sialic acid and nearby parts of the α2,6-linked galactoside establish contacts with the protein. **c** Binding of **2** by NL03 HA. The entire poly-LacNAc chain establishes contacts with the protein. **d** Comparison of glycan binding modes for NL91 and NL03 HA. A change in the C5-C6 torsion angle adjacent to the Neu5Ac-Gal glycosidic linkage facilitates a reorientation of the underling glycan chain for extended interaction with NL03 HA.

the Gal in the central LacNAc unit. Finally, the Y159F mutation produces a weaker CH-π acceptor[32].

Mutational reversibility has been observed in HAs[34], and for example in Sing16 (2016) residues 225 and 159 reverted back to D and Y, respectively, as in NL03 HA, and residue 193 back to F, as in NL09 HA. MD simulations confirmed that the key hydrogen bond between D225 and Gal-2 has been regained (Supplementary Fig. 10e) while the bulky side chain of F193 provokes steric clashes with the glycan chain as in NL09 HA. Finally, Y159 re-established strong CH-π interactions with Gal-6. Accordingly, NMR experiments showed strong $^1$H[$^{13}$C]-STD signals for the H3, H4, and H5 in Gal-6 as observed for NL03 HA.

To further assess the key contribution of Y159 for receptor binding, we produced a mutant of the Sing16 HA variant in which this tyrosine moiety is replaced by alanine, thereby abrogating the CH-π interaction. The NMR experiments showed substantial weaker $^1$H[$^{13}$C]-STD signals for Gal-6 (Fig. 7c, d) indicating much less intimate contacts. These results underpin that several residues cooperate in glycan binding.

## Discussion

Human influenza A viruses have a remarkable ability to evolve and evade neutralization by antibodies elicited by prior infections or vaccination[36–39]. This antigenic drift is mainly caused by mutations of amino acid in the globular head of HA where binding occurs with sialic acid-containing receptors of host cells. As a result, receptor binding modes of IAVs usually co-evolve by orchestrated mutations of several amino acids that allow immune evasion but maintain glycan binding capabilities[33]. A/H3N2 viruses, which are the leading cause of severe seasonal influenza illness[7], exhibit a particularly fast antigenic drift and have evolved to recognize sialosides presented on extended LacNAc moieties. Predicting influenza evolution requires an understanding, at a molecular level, how mutational changes driven by immune pressure shape glycan recognition. X-ray crystallography has provided key understandings of the recognition of sialosides by HAs[14,33,39,40]. Implementing such an approach for complex glycans such as **2** is, however, very difficult due to glycan flexible that results in diffuse electron densities complicating refinement of glycan structures within the

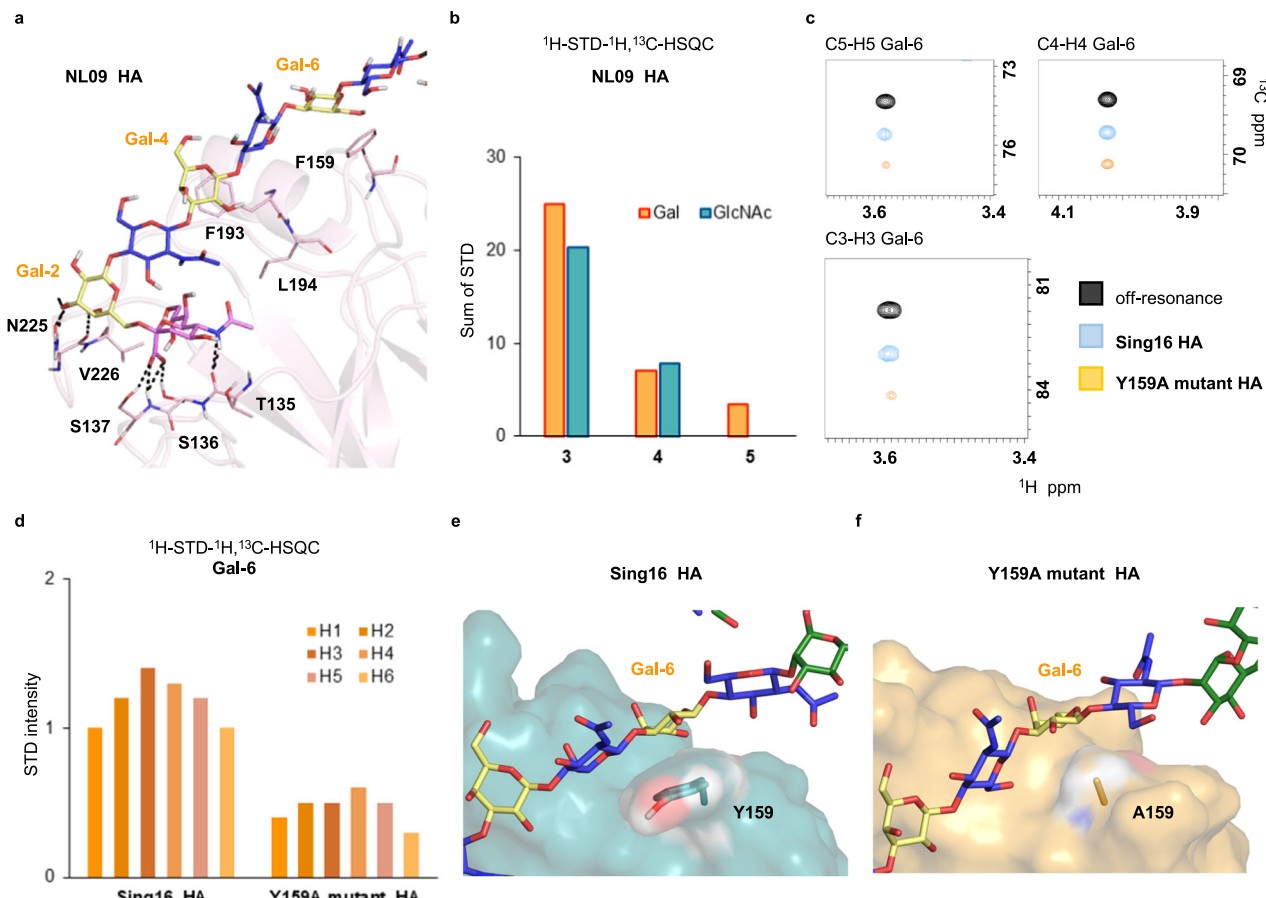

**Fig. 7 | Glycan binding by additional HA variants. a, b** Glycan binding by NL09 HA. MD derived complex structure with **2** (**a**), and ¹H[¹³C]-STD intensities from the 2D ¹H-STD-¹H,¹³C-HSQC spectrum of **3**–**5** (**b**). **c** Comparison of ¹H[¹³C]-STD signals of **5** in the presence of Sing16 HA and its Y159A HA mutant. The STD spectra are offset, in the vertical ¹³C dimension, to allow a comparison of their intensities. Note the significantly weaker STD signals for mutant Sing16 Y159A HA relative to the wild type. **d** ¹H-STD intensities for the ¹³C-labeled galactoside residue in **5**. **e, f** Model of glycan-HA interactions in Sing16 HA (**e**) and its Y159A HA mutant (**f**). The absence of an aromatic moiety in the Y159A mutant precludes the key CH-π interaction with Gal-6.

receptor binding site[41]. NMR signals reflect population-weighted averages of component conformers and binding modes under physiological conditions[42], and is recognized as a versatile technique to study glycans and glycan-receptor interactions with atomic resolution. Various experiments can be considered[43–47] and among these STD NMR is an attractive solution-based approach that can reveal which protons of a ligand interact with a protein and in combination with computational approaches, can provide models of glycan-protein complexes[48,49]. The implementation of such an approach for complex glycans is, however, complicated by NMR proton degeneracy. Recently, the binding of a symmetric sialylated-di-LacNAc containing *N*-glycan to an H3 protein was accomplished using an unnatural lanthanide tag to break the NMR signal degeneracy[50]. In this study, we addressed the problem of degeneracy by chemically synthesizing complex *N*-glycans having at specific positions ¹³C labeled monosaccharides without the need to introduce an artificial probe that may interfere in binding. To obtain the targeted isotopomers, we employed an *N*-glycan isolated from a natural source as starting material, recombinant human glycosyltransferases, and ¹³C labeled sugar nucleotide donors. Arm-selective extension of one of the arms of a symmetrical *N*-glycan starting material was possible by exploiting inherent branch selectivities of glycosyltransferases and glycosidases combined with modifications that temporarily block one of the arms from further extension[51]. The use of the synthetic isotopomers in 2D ¹H-STD-¹H,¹³C-HSQC experiments made it possible to unravel the contribution of individual monosaccharides of an extended poly-LacNAc

chain of an *N*-glycan for binding to evolutionary distinct HAs. In the case of the HA of the original pandemic A/H3N2 (HK68), only the sialoside engages with the protein and no contribution of the poly-LacNAc chain for binding was detected. For later strains, mutations close to the glycan binding site enabled interactions with the LacNAc chain. For the HA of the evolutionarily early NL91, these interactions are mostly limited to the sialic acid-linked galactoside while the more contemporary NL03 HA recognized the entire tri-LacNAc chain. The latter was confirmed by a fluorine-containing sialyl-LacNAc derivative as NMR probe that provided relative binding affinities and demonstrated additional contributions of the extended LacNAc chain for binding. The importance of an extended LacNAc moiety for binding of NL03 was also demonstrated by glycan microarray in which only a 2,6-sialoside presented at a tri-LacNAc moiety gave strong responsiveness. Thus, NMR solution-based binding studies as well as a binding assay in which glycans are immobilized on surface demonstrated the importance of the extended LacNAc moiety for binding of A/H3N2 viruses of the 3 C2 clade such as NL03. Previous studies have shown that fowl erythrocytes do not display sialosides on extended LacNAc moieties and accordingly NL03 cannot agglutinate these cells[16]. These viruses also propagate poorly in MDCK cells which also do not express these receptors[52,53] Respiratory tissues abundantly display, however, *N*-glycans having extended LacNAc moieties modified by sialosides making these appropriate receptors for IAV infection. It can, however, not be excluded that a high density of low-affinity receptor may also contribute to infection[19].

Combination of the NMR data with molecular modeling revealed how specific residues along the extended binding site of HA synergistically operate in glycan recognition. In particular, a G225D mutation changes the glycan orientation and moves the LacNAc chain closer to the protein surface. A distal mutation to the receptor binding site in NL03 HA reorient the Y159 side chain to face the Gal-6 making it possible to establish CH-π interactions. Mutagenesis experiments confirmed the important of the Y159 side chain for interacting with the extended LacNAc chain.

IAV have a remarkable ability to evolve and evade neutralization by antibodies elicited by prior infections or vaccinations. Understanding this evolution at a molecular level is important for vaccine design and the development predictive models for the evolution of IAV variants. Here we demonstrate for, the first time, that H3N2 viruses of the 3C.2 clade such as NL03 have evolved to make direct interactions with 2,6-sialoside presented on an extended LacNAc moiety resulting in a much higher affinity in binding.

## Methods

### Enzymatic synthesis

**Materials and general protocols.** Glycosyltransferase were expressed as previously described (St6Gal1, Fut1, St3Gal4, B4Galt1, and B3Gnt2)[54]. Galactosidase of *E. coli* was purchased from Sigma–Aldrich [Cat# G5635] α1,2-Fucosidase [Cat# P0724S] and α2,3-Neuraminidase [Cat# P0743] were purchased from New England BioLab. Alkaline phosphatase (FastAP) was purchased from Thermo Scientific [Cat# EF0651]. Uridine 5′-diphosphogalactose (UDP-Gal), uridine 5′-diphospho-N-acetyl-glucosamine (UDP-GlcNAc) and cytidine-5′-monophospho-N-acetylneuraminic acid (CMP-Neu5Ac) were obtained from Roche Diagnostics [UDP-Gal: Cat# 07703562103; UDP-GlcNAc: Cat# 06369855103; CMP-Neu5Ac: Cat# 05974003103]. $^{13}C$ labeled Uridine 5′-diphosphogalactose (UDP-Gal) and uridine 5′-diphospho-N-acetyl-glucosamine (UDP-GlcNAc) were obtained from Omicron [UDP-GlcNAc C1-C6: Cat# NTS-009; UDP-Gal C1-C6: Cat# NTS-005]. Reaction mixtures were purified by a size exclusion Biogel (P2) from BioRad in Econo glass columns (0.7 × 30 cm/1.5 × 30 cm/1.5 × 50 cm) coupled to a BioFrac fraction collector (BioRad). Carbohydrate-containing fractions were detected by thin-layer chromatography and an appropriate staining reagent (15 mL AcOH and 3.5 mL p-anisaldehyde in 350 mL EtOH and 50 mL $H_2SO_4$). Final products were purified by high-performance liquid chromatography (HPLC) using an XBridge HILIC column (10 mm (∅) × 250 mm (l), 5 μm particle size) on a semi-preparative liquid chromatography system from Shimadzu (LC-20AT, SIL-20A, CBM-20A, SPD-20A, FRC-10A). The purification was performed using 10 mM $NH_4HCO_3$ in 10% $H_2O$ in MeCN (buffer B) and 10 mM $NH_4HCO_3$ in 100% $H_2O$ (buffer A). The progress of the reactions was monitored on a Shimadzu liquid chromatography–mass spectrometry (LC–MS) (system controller: SCL10A-VP; HPLC pumps: LC10AD-VP; injector: SIL10AD-VP) using a ZIC HILIC column (ZeQuant, PEEK coated guard HPLC column, 3.5 μm particle size, 20 × 2.1 mm). The LC system was attached to a Bruker Daltonics microTOF-Q mass spectrometer.

### General procedures for synthesis

**Acid-mediated hydrolysis of Neu5Ac.** The substrate was dissolved in an aqueous solution of acetic acid (2 M) and kept at 65 °C for 24 h. The solvents were removed under an $N_2$ flow and the resulting mixture was applied to size exclusion chromatography. Carbohydrate-containing fractions were lyophilized and used without further purification.

**Installation of α2,6-linked Neu5Ac using ST6Gal1.** Acceptor and CMP-Neu5Ac (1.5 eq) were dissolved in a Tris buffer (50 mM, pH 7.3, 0.1 wt% BSA) to obtain an acceptor concentration of 2 mM. ST6Gal1 (42 μg per μmol acceptor) was added to the mixture and the resulting reaction mixtures was incubated overnight at 37 °C with gentle

shaking. The progress of the reaction was monitored by LC–MS. In case of incomplete conversion, additional ST6Gal1 (20 μg per μmol acceptor) was added and the reaction mixture was incubated at 37 °C for an additional 24 h. After completion of the reaction, the enzyme was removed by spin filtration using a 10 kDa cut-off filter, the filtrate was lyophilized and applied to size exclusion chromatography. Carbohydrate-containing fractions were collected, concentrated by freeze drying, and either used for further modification or further purification by HPLC.

**Installation of α1,2-linked fucoside using FuT1.** Acceptor and GDP-Fuc (1.3 eq) were dissolved in a Tris buffer (50 mM, pH 7.3, 0.1 wt% BSA) containing $MnCl_2$ (10 mM) to a final acceptor concentration of 5 mM. FuT1 (7 μg per μmol acceptor) were added and the reaction mixture was incubated at 37 °C for 72 h with gentle shaking. The progress of the reaction was monitored by LC–MS.

**Installation of β1,3-linked GlcNAc using B3GnT2.** Acceptor and UDP-GlcNAc (1.5 eq) were dissolved in a HEPES buffer (50 mM, pH 9.6, 0.1 wt % BSA) containing DTT (1 mM) and $MnCl_2$ (20 mM) to obtain a concentration of 5 mM. B3GnT2 (30 μg per μmol acceptor) and CIAP (1 u/μL, 1 u per μmol of added nucleotide) were added to the mixture and then incubated overnight at 37 °C with gentle shaking. The progress of the reaction was monitored by LC–MS. In case of incomplete conversion, additional UDP-GlcNAc (0.5 eq), CIAP (1 u/μL, 1 u per μmol of added nucleotide), and B3GnT2 (15 μg per μmol acceptor) were added and the reaction mixture incubated at 37 °C for an additional 24 h. The reaction mixture was lyophilized and applied to size exclusion chromatography. Carbohydrate-containing fractions were lyophilized and used without further purification.

**Installation of β1,4-linked galactoside using B4GalT1.** Acceptor and UDP-Gal (1.5 eq) were dissolved in a Tris buffer (50 mM, pH 7.3, 0.1 wt% BSA) containing $MnCl_2$ (20 mM) to obtain a concentration of 5 mM. B4GalT1 (20 μg per μmol acceptor) and CIAP (1 u/μL, 1 u per μmol of added nucleotide) were added and the resulting reaction mixture was incubated for 18 h at 37 °C with gentle shaking. The progress of the reaction was monitored by LC–MS. In case of incomplete conversion, additional UDP-Gal (0.5 eq), CIAP (1 u/μL, 1 u per μmol of added nucleotide), and B4GalT1 (10 μg per μmol acceptor) were added and the reaction mixture incubated at 37 °C for a further 24 h. The reaction mixture was lyophilized and applied to size exclusion chromatography. Carbohydrate-containing fractions were lyophilized.

**Fucoside hydrolysis.** The fucosylated glycan was dissolved into 50 mM sodium acetate buffer at pH 5.5, 5 mM $CaCl_2$, and treated with the broad specific α1-2,3,4,6 fucosidase (80 units). The reaction mixture was incubated at 37 °C with gentle shaking. The progress of the reaction was monitored by LC–MS.

**Galactoside hydrolysis.** The galacto-containing glycan was dissolved in 50 mM TRIS buffer at pH 7.3, 15 mM $MgCl_2$ and treated with galactosidase (20 units). The reaction mixture was incubated at 37 °C with gentle shaking. The progress of the reaction was monitored by LC–MS.

**Installation of α2,3-linked Neu5Ac using ST3Gal4.** Acceptor and CMP-Neu5Ac (1.5 eq) were dissolved in a Tris buffer 50 mM, pH 7.5, 1% v/v BSA (from stock solution 1 mg/mL), and 1% v/v CIAP (from stock solution 20 U/μL) to a final acceptor concentration of 5 mM. ST3Gal4 (42 μg per μmol acceptor) was added and the reaction mixture was incubated overnight at 37 °C with gentle shaking. The progress of the reaction was monitored by LC–MS.

**Enzymatic hydrolysis of α2,3-linked Neu5Ac.** The α2,3-sialylated glycan was dissolved into 50 mM sodium acetate buffer at pH 5.5, 5 mM

CaCl$_2$, and treated with the high specific α2,3- Neuraminidase S (160,000 U/mg). The reaction mixture was incubated at 37 °C with gentle shaking. The progress of the reaction was monitored by LC–MS.

## HA expression

Recombinant trimeric IAV hemagglutinin proteins open reading frames were cloned into the pCD5 expression vector as described previously[55], in frame with a GCN4 trimerization motif (KQIED-KIEEIESKQKKIENEIARIKK), a superfolder GFP[56] and the Twin-Strep-tag (WSHPQFEKGGGSGGGSWSHPQFEK); (IBA, Germany). The open reading frames of the HAs of A/Hong-Kong/1/1968 H3 (AFG71887.1), A/Netherlands/816/1991 H3 (EPI_ISL_114608), A/Netherlands/109/2003 H3 (EPI_ISL_113016), A/Netherlands/761/2009 H3 (EPI_ISL_1107270), A/Singapore/INFH-16-0019/2016 H3 (3 C.2a) (QQY97257.1), were synthesized and codon optimized by GenScript.

The trimeric HAs were expressed in HEK293T (CRL-11268) and HEK 293 S GNTI(-) (ATCC CRL-3022) cells with polyethyleneimine I (PEI) in a 1:8 ratio (μg DNA:μg PEI) for the HAs as previously described[16]. The transfection mix was replaced after 6 h by 293 SFM II suspension medium (Invitrogen, 11686029), supplemented with sodium bicarbonate (3.7 g/L), Primatone RL-UF (3.0 g/L, Kerry, NY, USA), glucose (2.0 g/L), glutaMAX (1%, Gibco), valproic acid (0.4 g/L) and DMSO (1.5%). Culture supernatants were harvested 5 days post-transfection and the HA0 was purified with sepharose strep-tactin beads (IBA Life Sciences, Germany) according to the manufacturer's instructions.

## NMR sample preparation and analysis

Proteins and ligands were dissolved in a deuterated TRIS-d$_{11}$ 50 mM buffer pD 7.8 in D$_2$O. Shigemi NMR tubes with a diameter of 3 mm were used. $^1$H-STD and $^1$H,$^{13}$C-STD HSQC experiments were acquired with a HA proteinsconcentration of 5 μM and a ligand at 300 μM (ratio 1:60). STD experiments were acquired on a 800 MHz BRUKER AVANCE III spectrometer, equipped with a TCI cryo-probe with z-gradient coil and TopSpin 3.2.7 (BRUKER) software was employed for data acquisition and processing. The temperature was set to 293 K. The 2D STD $^1$H,$^{13}$C-HSQC experiment was previously reported[21]. Briefly, the $^1$H,$^1$H-STD module was implemented with saturation by a train of 4 × 90° PC9_4 shaped pulses (with 1 ms separation) during d9 = 2 s, applied at the methyl $^1$H peak (0.84 ppm) with the non-saturation cut-offset at 1.18 ppm (i.e. distance Δν = 0.34 ppm), resulting in a PC9_4 pulse length of 33 ms at 800 MHz field strength. Protein saturation was alternated with off-resonant irradiation (at −25 ppm) in successive scans. The STD spectrum was then constructed by simple subtraction of both 2D $^1$H,$^{13}$C-HSQC spectra. Blank 2D STD-$^1$H,$^{13}$C-HSQC experiment of the HA glycoproteins alone and glycans **3**–**5** were acquired as control.

$^{19}$F CPMG NMR experiments were acquired on a 600 MHz spectrometer equipped with a Bruker selective $^{19}$F–$^1$H decoupling (SEF) probe at 298 K, using samples containing the protein at the concentration of 10 μM, and the F-SLN **6** at 125 μM. Competitor glycans **7** and **2** were successively added to the sample. The standard CPMG Bruker pulse sequence was modified as described[57]. Twenty-four (24) points were acquired with total echo times from 8 to 5200 ms, with τ = 2 ms. Data were analyzed with the T$_1$T$_2$ relaxation module of Topspin 3.5.

## In silico studies

**MD simulations.** When available, starting poses of the HAs were derived from X-ray crystal structures: HK68, A/HK/1/ 1968 H3N2 influenza virus hemagglutinin in complex with 6′-SLNLN (pdb code 6TZB)[33]. NL03, A/Wy/3/03 influenza virus hemagglutinin in complex with 6′- SLN (pdb code 6BKR)[34]. The starting poses of NL91, NL09, Sing16, and Sing16 (Y159A), were generated by superimposition of the model-derived structure [http:// 3dflu.cent.uw.edu.pl/index.html][58] onto the X-ray crystal structure of the closest variant. The 3D

coordinates of glycan **2** were generated by using the carbohydrate builder GLYCAM-web site [http://glycam.org]. The glycosidic torsion angles of the monosaccharides were maintained as observed by X-ray crystallography, while those not resolved were defined according to the lower energy values predicted by the GLYCAM-web modeling tool. The resulting poses were used as starting points for molecular dynamics (MD) simulations. The MD simulations were performed using the Amber16 program4 with the protein.ff14SB, the GLYCAM_06j-1, and the water.tip3p force fields parameters[59,60]. Next, the starting 3D geometries were placed into a 10 Å octahedral box of explicit TIP3P waters, and counter ions were added to maintain electro-neutrality. Two consecutive minimization steps were performed involving (1) only the water molecules and ions and (2) the whole system with a higher number of cycles, using the steepest descent algorithm. The system was subjected to two rapid molecular dynamic simulations (heating and equilibration). The equilibrated structures were the starting points for a final MD simulation at constant temperature (300 K) and pressure (1 atm). 1 μs molecular dynamic simulations without constraints were recorded, using an NPT ensemble with periodic boundary conditions, a cut-off of 10 Å, and the particle mesh Ewald method. A total of 500,000,000 molecular dynamics steps were run with a time step of 1 fs per step. Coordinates and energy values were recorded every 500,000 steps (500 ps) for a total of 1000 MD models. The detailed analysis of the H-bond and CH-π interactions was performed along the MD trajectory using the cpptraj module included in Amber-Tools 16 package.

## Glycan microarray binding of H3N2 viruses, HA proteins, and lectins

Virus isolates (25 μL) were diluted with PBS-T (PBS + 0.1% Tween, 25 μL) and applied to the array surface in the presence of oseltamivir (200 nM) in a humidified chamber for 1 h, followed by successive rinsing with PBS-T (PBS + 0.1% Tween), PBS, and deionized water (2×) and dried by centrifugation. The virus-bound slide was incubated for 1 h with the CR8020 A/H3N2 influenza hemagglutinin stem-specific antibody (made in-house, 100 μL, 5 μg/mL in PBS-T) and washed according to previous washing procedure. A secondary goat-anti-human Alexa 648 antibody (Thermo A-21445, 100 μL, 2 μg/mL in PBS-T) was applied, incubated for 1 h in a humidified chamber, and washed again as described above. The control lectins containing a biotin tag were visualized with Streptavidin-AlexaFluor555. Slides were dried by centrifugation after the washing step and scanned immediately. HAs were pre-complexed with human anti-streptag (made in-house) and goat-anti-human Alexa 555 (Thermo A-21433) in a 4:2:1 molar ratio, respectively in 50 μL PBS-T on ice for 15 min. The mixtures were added to the subarrays for 90 min in a humidified chamber. Wash steps after each incubation (e.g., enzyme treatment, HA, or antibody incubation) involved six successive washes of the whole slides with either twice PBS-T, twice PBS, and twice deionized water. The arrays were dried by centrifugation and immediately scanned as described previously[20]. Processing of the six replicates was performed by removing the highest and lowest replicates and subsequently calculating the mean value and standard deviation over the four remaining replicates.

## Reporting summary

Further information on research design is available in the Nature Portfolio Reporting Summary linked to this article.

## Data availability

The authors declare that the data supporting the findings of this study are available within the paper and its supplementary information files. Source data are provided with this paper. The sequence and 3D structures of HA of HK68 and NL03 were derived from https://www.rcsb.org/ under the access code 6TZB and 6BKR, respectively. Source data are provided with this paper.

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

## Acknowledgements
Research reported in this publication was supported by the Human Frontier Science Program Organization (HFSP) grant LT000747/2018-C (to L.U.), an ERC starting grant (802780) from the European Commission (to R.P.d.V.), and the National Institute of Allergy and Infectious Diseases of the National Institutes of Health under Award Number R01 AI165692 (to G.J.B). The content is solely the responsibility of the authors and does not necessarily represent the official views of the National Institutes of Health.

## Author contributions
L.U., A.N.A.A, G.P.B., E.U., R.L., F.B., R.W., Y.L., S.M., L.L., M.G.R., and I.A.B. performed the experiments. L.U., P.V., T.D., A.A., and R.P.V. analyzed the data and L.U., A.A., and R.P.V. interpreted the results. L.U., R.P.V., and G.J.B. wrote the manuscript. L.U., R.P.V., and G.J.B. supervised the research project. All authors have given approval to the final version of the manuscript.

## Competing interests
The authors declare no competing interests.
