## [Peer Review File · Nature Communications]

Probing Altered Receptor Specificities of Antigenically Drifting Human H3N2 Viruses by Chemoenzymatic Synthesis, NMR and ModelingREVIEWER COMMENTS

Reviewer #1 (Remarks to the Author):

This is certainly an elegant piece of work showing how the combination of enzymatic synthesis with selective introduction of ¹³C labelled sugar residues can improve the resolution in the level of detail about the molecular recognition of complex glycan molecules by protein receptor. All of that in the very important context of Influenza A virus evolution. The work is pretty solid, sound, and the NMR approaches have been carried out in a very smart way to be able to gain the desired information about the intermolecular interactions. The conclusions are strongly supported by the experimental evidences. All in all, I consider the merit of the work is worthy for publication in Nature Communications.

Nonetheless, I would like to appeal to the authors to clarify a few points:

- Regarding the competition experiments monitored by ¹⁹F NMR: to my opinion, although data are clearly compatible with a larger affinity for NL03, the competition experiments with 7 and 2 are not convincing for HK68 and NL91. For equal affinity of 6 and 7, an equimolar addition of one should reduce the impact of binding on R2 to half of the starting experiment. This is only evident for binding of 2 to NL03. Authors should clarify that, as it seems that the addition of two fluorine atoms are increasing significantly the affinity of the sialyl-LacNAc, and the natural ligand basically is not able to displace it from the binding pocket. Cannot the displacement be additionally monitored by STD NMR focusing on singular ¹H signals of the fluorinated ligand (e.g. HCF₂-)?
- Regarding the MD simulations, on page 11 (2nd paragraph) it is stated: Beyond the sialic acid residue, the MD simulation did not reveal stable intermolecular interactions and throughout the MD trajectory the poly-LacNAc chain was flexible and mainly solvent exposed (Supplementary Fig. 8). However, looking at that figure, the solvent exposure of the poly-LacNAc is far from evident. Could it be that the chosen molecular orientation in the figure is not appropriate to reveal that aspect? This should be modified to clarify this issue.

Very minor issues:

- Page 4: "magnetic saturation" does not seem to be a proper term; please substitute by simply "saturation", or "magnetization saturation"
- Page 7 (end of 2nd paragraph): "...is due to additional recognition of the Gal and GlcNAc residues absent in HK68". Not properly expressed, Gal and GlcNAc are not absent in any case (I know authors refer to additional recognition, yet, as stated, not clearly expressed)
- Figure 2a: I would recommend rather a tilted stacked plot, as in the current display it is impossible to distinguish the STD outcomes of the acetamide methyl groups.
- Figure 3a: quality of the figure very poor. Please improve it.
- Page 10: "...during evolution of HA evolution...", repetition
- Page 11 (3rd paragraph): please use "not observed" rather than "unobservable". Subtle difference but certainly important.
- Page 11 (3rd paragraph): "van der Waals", instead of "Van der Waals"
- Page 12: what is RBS?

Reviewer #2 (Remarks to the Author):

This manuscript addresses the interesting question of changes in specificity of the hemagglutinin (HA) on the surface of influenza A virus for different types of sialic acid-containing oligosaccharides or glycans. The evidence indicates that there is a change over the prior decades in specificity of IAV for sialic acid on extended glycans containing multiple disaccharide repeats of LacNAc (Gal-GlcNAc). Here the authors address this issue with synthetic glycans containing ¹³C-labeled residues in specific positions for use in NMR experiments that allow them to explore protein interactions (through the recombinant hemagglutinins) with these tagged glycans. The results are

interpreted to indicate that there is an enlarged glycan-binding domain with the HA and that contacts occur with the sialic acid residue and residues within the LacNAc repeats. Affinities are measured using a fluorine-containing sialoside. The NMR experimental results were combined with molecular modeling studies to provide predicted intermolecular interactions of protein and glycans. Overall, these results are novel and interesting and provide context for understanding the change in specificity of the HA over time within the H3N2 IAVs. The studies are limited to structural and biochemical interpretations and approaches and the nature and importance of such ligands described here to natural infections of cells by these viruses is not clear.

As the glycans used are relatively large and asymmetrical, the question arises as to the role of multivalency when the glycan branches are all sialylated and/or extended. Does this increase specificity, or is there 'crowding' between glycans that may actually reduce binding?

How important is the sialic acid residue compared to the extended LacNAc sequence in the overall affinity and specificity, and how important is the 2,6-linkage versus the 2,3-linkage in this context?

Did the authors test the specific differences in 2,3-linkages versus 2,6-linkages in the context of these extended glycans?

The authors make the statement that "...while the more contemporary NL03 HA recognised almost the entire poly-LacNAc.", but the evidence for that is unclear, as such chains may have different lengths, etc., so the authors should be specific in this context.

The alternate isomers of those glycans in Fig. 1 are not tested for their interactions.

All the synthetic glycans shown in Fig. 1 have sialic acid on one arm of the N-linked glycan, and the other arm is not modified. In nature, these glycans may be relatively minor, and branched and extensively modified glycans may be very common. Overall, in this context, this reviewer is not aware of the evidence for such structures as shown in Fig. 1 being definitively identified in human tissues as the required ligands for infections of cells by recent strains of IAVs. It seems that the evidence is mainly based on microarray studies, but not specific biological, biochemical, and glycomic studies on the glycans required in tissues for infection. In this context, even low affinity ligands, if abundant might function. For the average reader the context of these synthetic materials that are presumably modeled on the native ligands for IAVs in human tissues should be made clear if possible. Otherwise, the studies would appear to be limited in their biological impact.

In that regard it is not clear to this reviewer as to the specific dependency on chain length of the LacNAc repeats for the different hemagglutinins, i.e. changes in affinity and specificity versus interactions with specific residues.

The chemical binding affinities of each of the glycans described in this study are not defined for each of the hemagglutinins tested.

The descriptions of the changes in specificity of the hemagglutinins over the past decades have been made in numerous publications, so the key insights here appear to be the evidence that the residues with the LacNAc repeats make contact with the protein, rather than just providing an extended version of a sialosides, and while interesting, this does limit the impact of these overall studies.

Minor: The authors do not clearly define STD as saturation transfer difference.

Reviewer #3 (Remarks to the Author):

The manuscript by Unione et al. is designed to give new insights into the interaction between influenza H3 HA and its receptor in humans, particularly with respect to antigenic drift observed in receptor binding domain of H3 HA. Specifically, the authors have synthesized receptor analogs (13C labeled N-glycans and 19F labeled sialoside), characterized the interaction of recombinant HA

with receptor analogs by NMR spectroscopy, and modeled the interaction based on computational and mutagenesis studies.

The work significantly advances the previous work of the group using glycan arrays and computer modeling on the same system (Broszeit et al., 2021, reference 17 of the manuscript). In general the manuscript is well written, the data are of high quality, and the experimental results justify the author's conclusions. Importantly, influenza poses significant threat to human health and a better understanding of the driving forces of antigenic drift may lead to better vaccination and/or therapeutics. Nonetheless there are a few weaknesses to be addressed.

Weaknesses:

1. For the 19F experiments, the authors should explain why they did not label the more complex glycans with 19F, which would allow more accurate characterization of binding affinities. Furthermore, why not measure binding affinity for the 19F-labeled compound (i.e. titrate the substrate)? In addition, the authors should explain their rationale behind using 1.0 or 0.4 equivalents of complex glycans in the competition assay. Finally, it would be nice to see the 19F spectra in the Supplementary Material (and were changes in chemical shift observed?).
2. Based on Figure 5c, the authors observe significant changes to the 13C chemical shift of the Gal in the presence of the HA variant and site-directed mutant. The authors need to comment on this.
3. In the Methods, the authors should to comment whether the recombinant HA used in the experiments was processed (i.e. was the HA1-HA2 site cleaved).

Reviewer #4 (Remarks to the Author):

In this study, Unione et al. has synthesized a series of 13C labelled N-glycans to illustrate the evolution of human H3N2 HA receptor binding mode using NMR, by focusing on three H3N2 strains (HK68, NL91, and NL03). Additionally, the authors performed molecular dynamic (MD) simulations and mutagenesis to support their conclusions. This study has identified several key HA residues that are responsible for the evolution of receptor binding mode. While this study is technically well performed, the evolution of H3N2 receptor binding mode has previously been shown by X-ray crystallography (ref #30 and ref #32), which hamper the novelty of the present study. Also, Fernandez de Toro et al. has reported the use of STD NMR to study HA-receptor interaction in 2018 (PMID: 30238596) but is not cited in this study. That being said, this study provides some previously unknown details about the evolution of H3N2 receptor binding mode (e.g. interaction between Y159 and Gal-6). Moreover, the NMR result demonstrates the importance of extended LacNAc moieties in the receptor binding of more recent H3N2 strains, confirming the previous findings from a glycan microarray study (ref #16) and represents a major contribution to the understanding of HA-receptor interaction.

Major comments:

1. Based on MD simulation, S145K has a compensatory effect for N137Y. This interesting finding needs to be experimentally validated by mutagenesis.
2. To the best of my knowledge, Fernandez de Toro et al. first reported the use of STD NMR to study HA-receptor interaction in 2018 (PMID: 30238596). This study needs to be cited.
3. In the 2nd and 3rd paragraphs under section "Modelling of glycan-HA complexes", many key HA-receptor interactions in recent H3N2 strains have identified by MD simulation. The authors need to compare these interactions identified by MD simulation with those previously observed in crystal structures, including PDB 6BKR (A/Wyoming/3/2003 + 6'-SLN, ref #32), PDB 6BKT (A/Michigan/15/2014 + 6'-SLN, ref #30), and PDB 6AOV (A/Brisbane/10/2007 + 6'-SLN, PMID: 29059230).

Minor comments:

1. In the first paragraph of introduction, it might be clearer to rephrase "12-56,000 deaths" as

"12–56 thousand deaths"

2. "Indeed, the 2D ^1H -STD- ^1H , ^{13}C -HSQC experiment with the three isotopomers 3, 4, and 5 showed substantially weaker ^1H [^{13}C]-STD signals than observed for NL03 HA (Fig. 5a)." It should be 5b instead of 5a.

3. "MD simulation revealed that N145 cannot engage the sialic acid by hydrogen bonding interaction in the way K145 does for NL03 HA (Supplementary Fig. 7d)." N145 is not labeled in Supplementary Fig. 7d.

4. The authors should consider changing "H3Sing" to "Sing16", so that the abbreviation scheme for influenza strain names is consistent throughout the manuscript (e.g. HK68, NL91, and NL03).

5. In the first paragraph of discussion, "recognition of sialosides by Has" should be "recognition of sialosides by HAs".

We thank the reviewers for their supporting remarks and suggested improvements. The revised manuscript addresses unclarities. Furthermore, additional experiments have been performed including detailed binding studies using STD NMR to provide K_D values for sialosides having LacNAc moieties of varying length; glycan microarray data to demonstrate receptor specificities and binding studies using mutant HAs to demonstrate the importance of specific amino acids for binding. The new data reinforces the previously determined conclusions regarding receptor binding specificities of evolutionary distinct H3N2 viruses.

Reviewer #1 (Remarks to the Author):

This is certainly an elegant piece of work showing how the combination of enzymatic synthesis with selective introduction of ^{13}C labelled sugar residues can improve the resolution in the level of detail about the molecular recognition of complex glycan molecules by protein receptor. All of that in the very important context of Influenza A virus evolution. The work is pretty solid, sound, and the NMR approaches have been carried out in a very smart way to be able to gain the desired information about the intermolecular interactions. The conclusions are strongly supported by the experimental evidence. All in all, I consider the merit of the work is worthy for publication in Nature Communications.

Nonetheless, I would like to appeal to the authors to clarify a few points:

Remark 1.1. Regarding the competition experiments monitored by ^{19}F NMR: to my opinion, although data are clearly compatible with a larger affinity for NL03, the competition experiments with 7 and 2 are not convincing for HK68 and NL91. For equal affinity of 6 and 7, an equimolar addition of one should reduce the impact of binding on R2 to half of the starting experiment. This is only evident for binding of 2 to NL03. Authors should clarify that, as it seems that the addition of two fluorine atoms are increasing significantly the affinity of the sialyl-LacNAc, and the natural ligand basically is not able to displace it from the binding pocket.

Response 1.1. In the limit of fast exchange, the observed R2 ($R_{2,\text{obs}}$) is the weighted average of the relaxation rate in the free ($R_{2,\text{f}}$) and bound state ($R_{2,\text{b}}$) plus a term that accounts for line broadening, the exchange term (R_{ex}). Thus, the $R_{2,\text{obs}}$ is proportional to the fraction of bound (p_b) as well as to the fraction of free probe (p_f), according to the equation:

$$R_{2,\text{obs}} = p_f R_{2,\text{f}} + p_b R_{2,\text{b}} + R_{\text{ex}}$$

Based on previous research (*Nature*, 497, 392–396, 2013), the dissociation constant (K_D) for binding of hemagglutinin to sialyl N-acetyl-lactosamine is in the low mM to high μM range, which results in a high fraction of free probe (p_f). Therefore, a slight decay of $R_{2,\text{probe}}$ is expected when a competitor of similar affinity is used. Our aim was to demonstrate the suitability of compound 6 as probe. Under the afore mentioned conditions, the observed small decay of $R_{2,\text{probe}}$ is consistent with the 5% decay observed for HK68 and NL91. The same can be deduced for compound 7 for binding with NL03. We conclude that compounds 6 and 7 bind with similar relatively weak affinities at the same binding site of the HA proteins. On the other hand, a decay of *ca.* 30% in R2 was observed upon addition of 0.4 eq of 2 to 6 and NL03 which fits with dissociation constant of at least one order of magnitude stronger than that determined

for the sialylated mono-LacNAc derivatives **6** and **7**. To reinforce our analysis, we simulated the R2 decay for the different systems. The additional study have been integrated in SI section 4 “¹⁹F-T₂ relaxation”. The main text has been modified accordingly.

Remark 1.2. Cannot the displacement be additionally monitored by STD NMR focusing on singular ¹H signals of the fluorinated ligand (e.g. HCF₂-)?

Response 1.2. We thank the reviewer for the suggestion. R₂(¹⁹F) NMR experiments require a relatively low protein/ligand ratio (in this study approx. 1:12), whereas ¹H STD NMR experiments commonly require higher ratios (in this study approx. 1:60). We expanded our study by preparing two new ligands (compounds **8** and **9**) in suitable quantities for NMR experiments. ¹H-STD NMR experiments were performed to monitor the singular ¹H signals of compound **6** as function of the addition of increasing equivalents of competitors **7-9** (Fig. 1). In particular, the intensity of the unique ¹H-STD signal of the di-fluoro acetamide moiety was analyzed. The results of these additional experiments provided dissociation constant K_{Comp} for each ligand in complex with NL03. Consistently with our previous conclusions, we found that the mono- and the di-LacNAc containing compounds bind to the protein with K_{DS} of around 0.3-0.6 mM. A similar value was observed for the reference probe **6** (ca. 0.7 mM). Instead, a K_D of ca. 30 μM was observed for tri-LacNAc containing compound **9**. These results allowed for quantification of the contribution of the underlying tri-LacNAc chain to binding with NL03, whose interaction is at least order of magnitude stronger than that of the mono- and di-LacNAc-containing compounds. The resulting ¹H-STD spectra are reported in SI section 4 and have been integrated in the main text of the manuscript: “*¹H-STD competition experiments were performed to quantify the contribution of the extended LacNAc moiety to the K_D of NL03. Thus, the intensity of unique ¹H-STD signal of the di-fluoro acetamide moiety of **6** was monitored as function of different concentrations of competitors **7**, **8** and **9** (Fig. 1). Consistent with the ¹⁹F-NMR experiments, the mono- (**7**) and the di-LacNAc (**8**) containing sialosides bound to the protein with a K_D of ~600 μM, which is similar to that derived for the fluorinated compound **6**. The K_D of the tri-LacNAc containing sialoside **9** was determined to be 30 μM. These results further support that the underlying tri-LacNAc chain of **9** substantially contributes to binding of NL03, resulting in a 10-fold higher affinity compared to the mono- and di-LacNAc containing sialosides (The corresponding ¹H-STD spectra are reported in Supplementary Fig. 5c.)*”.

Remark 1.3. Regarding the MD simulations, on page 11 (2nd paragraph) it is stated: Beyond the sialic acid residue, the MD simulation did not reveal stable intermolecular interactions and throughout the MD trajectory the poly-LacNAc chain was flexible and mainly solvent exposed (Supplementary Fig. 8). However, looking at that figure, the solvent exposure of the poly-LacNAc is far from evident. Could it be that the chosen molecular orientation in the figure is not appropriate to reveal that aspect? This should be modified to clarify this issue.

Response 1.3. We agree that the chosen orientation does not clearly show the solvent exposed pose of the poly-LacNAc chain. We have changed the Supplementary fFig. 8, now SI Fig. 10, accordingly.

Very minor issues:

Remark. Page 4: “magnetic saturation” does not seem to be a proper term; please substitute by simply “saturation”, or “magnetization saturation”

Response. We thank the reviewer for indicating this mistake. We removed the word “magnetic” as suggested.

Remark. Page 7 (end of 2nd paragraph): “...is due to additional recognition of the Gal and GlcNAc residues absent in HK68”. Not properly expressed, Gal and GlcNAc are not absent in any case (I know authors refer to additional recognition, yet, as stated, not clearly expressed)

Response. We changed the sentence by: “is due to additional recognition of the Gal and GlcNAc residues which does not take place for HK68”.

Remark. Figure 2a: I would recommend rather a tilted stacked plot, as in the current display it is impossible to distinguish the STD outcomes of the acetamide methyl groups.

Response. We thank the reviewer for the suggestion. We have restyled the figure, including a zoom in of the acetyl $^1\text{H-NMR}$ signal region.

Remark. Figure 3a: quality of the figure very poor. Please improve it.

Response. The quality of the figure has been improved.

Remark. Page 10: “...during evolution of HA evolution...”, repetition

We thank the reviewer for indicating this mistake.

Remark. Page 11 (3rd paragraph): please use “not observed” rather than “unobservable”. Subtle difference but certainly important.

Response. We changed the text as suggested by the referee.

Remark. Page 11 (3rd paragraph): “van der Waals”, instead of “Van der Waals”

Response. We thank the reviewer for indicating this mistake, which has been corrected.

Remark. Page 12: what is RBS?

Response. RBS is receptor binding site. In the revised manuscript it is explicitly indicated as “receptor binding site (RBS)”.

Reviewer #2 (Remarks to the Author):

This manuscript addresses the interesting question of changes in specificity of the hemagglutinin (HA) on the surface of influenza A virus for different types of sialic acid-containing oligosaccharides or glycans. The evidence indicates that there is a change over the prior decades in specificity of IAV for sialic acid on extended glycans containing multiple disaccharide repeats of LacNAc (Gal-GlcNAc). Here the authors address this issue with synthetic glycans containing ^{13}C -labeled residues in specific positions for use in NMR experiments that allow them to explore protein interactions (through the recombinant hemagglutinins) with these tagged glycans. The results are interpreted to indicate that there is an enlarged glycan-binding domain with the HA and that contacts occur with the sialic acid residue and residues within the LacNAc repeats. Affinities are measured using a fluorine-containing sialoside. The NMR experimental results were combined with molecular modeling studies to provide predicted intermolecular interactions of protein and glycans. Overall, these

results are novel and interesting and provide context for understanding the change in specificity of the HA over time within the H3N2 IAVs. The studies are limited to structural and biochemical interpretations and approaches and the nature and importance of such ligands described here to natural infections of cells by these viruses is not clear.

Remark 2.1. As the glycans used are relatively large and asymmetrical, the question arises as to the role of multivalency when the glycan branches are all sialylated and/or extended. Does this increase specificity, or is there ‘crowding’ between glycans that may actually reduce binding?

Response 2.1. The surface of the influenza A viruses is covered with hemagglutinin (HA) proteins that are presented as trimers which can form multiple interactions with sialic acid-terminated glycans on the host cell surface. Multivalency is widely regarded as being important for the binding of influenza A virus to host cells. We have included glycan microarray binding studies in which multiple glycans are presented on a surface for binding with viruses. It includes compounds having α 2,6- and α 2,3-linked sialosides, as well as, none-sialylated compounds. It also includes compounds having an extended LacNAc moiety (see SI section 3, Fig. 4). The results show that sialic acid is essential for recognition. Evolutionary early HAs, such as HK68, can recognize α 2,3- as well as α 2,6-sialylated glycans. At a later stage in the evolution, such as for NL91, only recognize α 2,6 sialylated glycans are recognized without a dependence on an extended LacNAc chain. More recent viruses, such as NL03, only bind to α 2,6 sialylated glycans on an extended LacNAc chain. The following paragraph is included: “*First, we examined the receptor binding specificities of HK68, NL91 and NL03 by glycan microarray technology. Thus, a series of N-glycans was printed on NHS-activated microarray slides which was probed by several plant lectins for quality control (Supplementary Fig. 4). Next, whole viruses of HK68, NL91 and NL03 were exposed to the glycan microarray and binding was determined by using broadly neutralizing antibody CR8020 and an Alexa labelled goat-anti-human antibody (Supplementary Fig. 4). HK68 displays some residual avian-type receptor binding specificity and bound 2,3- as well as 2,6-sialosides. NL91 only bound human-type receptors including compound 1 and 2 that display the 2,6-linked sialoside at a mono- and tri-LacNAc moiety, respectively. The further evolved NL03 only bound compound 2. This virus prefers the sialoside at the extended LacNAc moiety to be displayed at the α 3-mannose antenna as the isomeric compound did not show binding. Mono- and di-sialylated structures gave similar responsiveness indicating that bidentate binding is not important for recognition. Control compounds lacking sialic acid did not bind.*”

The conclusion section has also been extended to address noted issues “*In the case of the HA of the original pandemic A/H3N2 (HK68), only the sialoside engages with the protein and no contribution of the poly-LacNAc chain for binding was detected. For later strains, mutations close to the glycan binding site enabled interactions with the LacNAc chain. For the HA of the evolutionarily early NL91, these interactions are mostly limited to the sialic acid-linked galactoside while the more contemporary NL03 HA recognised the entire tri-LacNAc chain. The latter was confirmed by a fluorine containing sialyl-LacNAc derivative as NMR probe that provided relative binding affinities and demonstrated additional contributions of the extended LacNAc chain for binding. The importance of an extended LacNAc moiety for binding of NL03 was also demonstrated by glycan microarray in which only a 2,6-sialoside presented at a tri-LacNAc moiety gave strong responsiveness. Thus, NMR solution-based binding studies as well as a binding assay in which glycans are immobilized on surface demonstrated the importance*

of the extended LacNAc moiety for binding of A/H3N2 viruses of the 3C2 clade such as NL03. Previous studies have shown that fowl erythrocytes do not display sialosides on extended LacNAc moieties and accordingly NL03 cannot agglutinate these cells¹⁷. These viruses also propagate poorly in MDCK cells which also do not express these receptors^{54,55}. Respiratory tissues abundantly display, however, N-glycans having extended LacNAc moieties modified by sialosides making these appropriate receptors for IAV infection. It can, however, not be excluded that a high density of low affinity receptor may also contribute to infection⁵⁶.”

Remark 2.2. How important is the sialic acid residue compared to the extended LacNAc sequence in the overall affinity and specificity, and how important is the 2,6-linkage versus the 2,3-linkage in this context? Did the authors test the specific differences in 2,3-linkages versus 2,6-linkages in the context of these extended glycans?

Response 2.2. As described above, these issues have been addressed by including glycan microarray data.

Remark 2.3. The authors make the statement that “...while the more contemporary NL03 HA recognised almost the entire poly-LacNAc..”, but the evidence for that is unclear, as such chains may have different lengths, etc., so the authors should be specific in this context.

Response 2.3. We have changed the sentence according to the suggestion of the referee: “...while the more contemporary NL03 HA recognised almost the entire poly-LacNAc” by “...while the more contemporary NL03 HA recognised almost the entire tri-LacNAc chain.”

Remark 2.4. The alternate isomers of those glycans in Fig. 1 are not tested for their interactions.

Response 2.4. ST6Gal1, which is the only enzyme that installs 2,6-sialosides at LacNAc, preferentially modifies the α 3-arm of N-glycans and therefore this series of compounds was prepared for the NMR experiments. The glycan microarray includes alternate isomers and it shown that the A/H3N2 viruses prefer the sialoside to be presented at the α 3-arm.

Remark 2.5. All the synthetic glycans shown in Fig. 1 have sialic acid on one arm of the N-linked glycan, and the other arm is not modified. In nature, these glycans may be relatively minor, and branched and extensively modified glycans may be very common. Overall, in this context, this reviewer is not aware of the evidence for such structures as shown in Fig. 1 being definitively identified in human tissues as the required ligands for infections of cells by recent strains of IAVs. It seems that the evidence is mainly based on microarray studies, but not specific biological, biochemical, and glycomic studies on the glycans required in tissues for infection. In this context, even low affinity ligands, if abundant might function. For the average reader the context of these synthetic materials that are presumably modeled on the native ligands for IAVs in human tissues should be made clear if possible. Otherwise, the studies would appear to be limited in their biological impact.

Response 2.5. We thank the reviewer for the comment. N-linked glycans having extended LacNAc moieties that are capped by sialosides have been documented on a human epithelial cell line (*Nat. Biotechnol.* 26, 107-113) and respiratory tissues (*Sci. Rep.* 10, 5320, 2020 and *PLoS Pathog.* 9, e1003223, 2013) and are expected to be receptors of IAVs. We have included the following paragraph to clarify this issue: “We have constructed a glycan microarray populated with bi-antennary N-glycans that more closely resemble structures expressed by human respiratory tissue¹⁷. It includes N-glycans having multiple consecutive LacNAc

repeating units that can be capped by sialic acid. Glycomic analysis has shown that these structures are abundantly expressed by a human epithelial cell line¹⁷ and respiratory tissues¹⁸ and are expected to be relevant receptors for IAVs.” The conclusion section provides a section that provides further support for these structures as receptor for IAVs: “Thus, NMR solution-based binding studies as well as a binding assay in which glycans are immobilized on surface demonstrated the importance of the extended LacNAc moiety for binding of A/H3N2 viruses of the 3C2 clade such as NL03. Previous studies have shown that fowl erythrocytes do not display sialosides on extended LacNAc moieties and accordingly NL03 cannot agglutinate these cells¹⁷. These viruses also propagate poorly in MDCK cells which also do not express these receptors^{54,55} Respiratory tissues abundantly display, however, N-glycans having extended LacNAc moieties modified by sialosides making these appropriate receptors for IAV infection. It can, however, not be excluded that a high density of low affinity receptor may also contribute to infection⁵⁶.”

Remark 2.6. In that regard it is not clear to this reviewer as to the specific dependency on chain length of the LacNAc repeats for the different hemagglutinins, i.e. changes in affinity and specificity versus interactions with specific residues. The chemical binding affinities of each of the glycans described in this study are not defined for each of the hemagglutinins tested.

Response 2.6. We thank the reviewer for raising this point. We have included additional NMR studies to measure affinities. As these studies required relatively large quantities of glycans, two new derivatives were prepared (compounds **8** and **9**) that are 2,6-sialosides presented on a di-LacNAc and tri-LacNAc chain, which together with mono-LacNAc derivative **7** were used in competition NMR experiments. It provided dissociation constant K_D for each ligand in complex with NL03. Consistently with our previous results, we found that the mono- and the di-LacNAc containing sialosides bind to the protein with a K_D of $\sim 600 \mu\text{M}$. A similar value was measured for the fluorinated compound **6**. On the other hand, a K_D of 30 mM was determined for the tri-LacNAc containing sialoside **9**. The combination of ¹H-STD-¹H,¹³C_HSQC experiments, modelling, and the derived dissociation constant demonstrate that the interaction of the tri-LacNAc chain with specific residues, such as D225, S193, and Y159 in NL03 accounts for 10-fold higher affinity than the mono- and di-LacNAc containing compounds.

Remark 2.7. The descriptions of the changes in specificity of the hemagglutinins over the past decades have been made in numerous publications, so the key insights here appear to be the evidence that the residues with the LacNAc repeats make contact with the protein, rather than just providing an extended version of a sialosides, and while interesting, this does limit the impact of these overall studies.

Response 2.7. Although previous studies had indicated that recent H3N2 viruses have altered receptor binding specificities, the contribution of the extended LacNAc chain for binding was not understood. Previous X-ray crystallography (ref. 35-36) and NMR study (ref 53) did not reveal the contribution of a tri-LacNAc moiety for binding, which is addressed in this study. The combined NMR, modelling and mutagenesis studies reveal amino acids critical for forming an extended binding site. The importance of these finding is further discussed in the conclusion section: “IAV have a remarkable ability to evolve and evade neutralization by antibodies elicited by prior infections or vaccinations. Understanding this evolution at a molecular level is important for vaccine design and the development predictive models for the

evolution of IAV variants. Here we demonstrate for, the first time, that H3N2 viruses of the 3C.2 clade such as NL03 have evolved to make direct interactions with 2,6-sialoside presented on an extended LacNAc moiety resulting in a much higher affinity in binding.”

Minor: The authors do not clearly define STD as saturation transfer difference.

We defined STD as saturation transfer difference in the introduction. Page 3, line 13.

Reviewer #3 (Remarks to the Author):

The manuscript by Unione et al. is designed to give new insights into the interaction between influenza H3 HA and its receptor in humans, particularly with respect to antigenic drift observed in receptor binding domain of H3 HA. Specifically, the authors have synthesized receptor analogs (13C labeled N-glycans and 19F labeled sialoside), characterized the interaction of recombinant HA with receptor analogs by NMR spectroscopy, and modeled the interaction based on computational and mutagenesis studies.

The work significantly advances the previous work of the group using glycan arrays and computer modeling on the same system (Broszeit et al., 2021, reference 17 of the manuscript). In general the manuscript is well written, the data are of high quality, and the experimental results justify the author’s conclusions. Importantly, influenza poses significant threat to human health and a better understanding of the driving forces of antigenic drift may lead to better vaccination and/or therapeutics. Nonetheless there are a few weaknesses to be addressed.

Remark 3.1. For the 19F experiments, the authors should explain why they did not label the more complex glycans with 19F, which would allow more accurate characterization of binding affinities. Furthermore, why not measure binding affinity for the 19F-labeled compound (i.e. titrate the substrate)? In addition, the authors should explain their rationale behind using 1.0 or 0.4 equivalents of complex glycans in the competition assay. Finally, it would be nice to see the 19F spectra in the Supplementary Material (and were changes in chemical shift observed?).

Response 3.1. We thank the reviewer for the comment. The synthesis of branched extended glycans with the ¹⁹F probe is challenging. We expanded our study by preparing five new ligands, di- (compound **8**), and tri- (compound **9**) LacNAc terminated with 2,6-sialosides, which, together with **7**, represent the receptors for the HAs, together with the fluorinated counterparts (**6**, **10**, and **11**). These molecules were used in ¹⁹F-R₂ and ¹H-STD NMR titration and competition experiments. The results of the additional experiments has provided dissociation constant (K_D) for each ligand in complex with HA of NL03 and NL91. Consistent with our previous findings, we found that the mono- and the di-LacNAc sialosides bind to the NL03 protein with a K_D of ~600 mM. A similar value was observed for the fluorinated compound **6**. On the other hand, a K_D of 30 mM was measured for the tri-LacNAc containing sialoside **9**. These results confirm the importance of the underlying tri-LacNAc chain for binding with NL03. In the case of NL91, a K_D of ca. 2 mM was measured for the mono-LacNAc compounds (**6** and **7**). Both ¹⁹F-R₂ and ¹H-STD NMR experiments demonstrated that the di- and tri-LacNAc containing sialosides weakly compete with the mono-LacNAc compounds, indicating similar affinities. The resulting ¹H-STD and ¹⁹F-NMR spectra are reported in SI section 4, and discussed in the main text of the manuscript. The synthesis of the fluorinated compounds is presented in SI section 2 together with the ¹⁹F NMR spectra.

Remark 3.2. Based on Figure 5c, the authors observe significant changes to the ¹³C chemical shift of the Gal in the presence of the HA variant and site-directed mutant. The authors need to comment on this.

Response 3.2. We apologise for failing to indicate more clearly that the STD spectra shown in Fig. 5c were deliberately offset (in the vertical ¹³C dimension) to allow a comparison of their intensities and are not the consequence of any binding induced signal shifting. We hope to have clarified this misconception by an addition made to the Fig. 5c caption.

Remark 3.3. In the Methods, the authors should to comment whether the recombinant HA used in the experiments was processed (i.e. was the HA1-HA2 site cleaved).

Response 3.3. We have added this information in the Methods. Page 21, line 1.

Reviewer #4 (Remarks to the Author):

In this study, Unione et al. has synthesized a series of ¹³C labelled N-glycans to illustrate the evolution of human H3N2 HA receptor binding mode using NMR, by focusing on three H3N2 strains (HK68, NL91, and NL03). Additionally, the authors performed molecular dynamic (MD) simulations and mutagenesis to support their conclusions. This study has identified several key HA residues that are responsible for the evolution of receptor binding mode. While this study is technically well performed, the evolution of H3N2 receptor binding mode has previously been shown by X-ray crystallography (ref #30 and ref #32), which hamper the novelty of the present study. Also, Fernandez de Toro et al. has reported the use of STD NMR to study HA-receptor interaction in 2018 (PMID: 30238596) but is not cited in this study. That being said, this study provides some previously unknown details about the evolution of H3N2 receptor binding mode (e.g. interaction between Y159 and Gal-6). Moreover, the NMR result demonstrates the importance of extended LacNAc moieties in the receptor binding of more recent H3N2 strains, confirming the previous findings from a glycan microarray study (ref #16) and represents a major contribution to the understanding of HA-receptor interaction.

Remark 4.1. Based on MD simulation, S145K has a compensatory effect for N137Y. This interesting finding needs to be experimentally validated by mutagenesis.

Response 4.1. We thank the Reviewer for raising this point. To validate the effect of amino acid at positions 137 and 145, we introduced K145S and N137Y in NL03 and examined their binding properties by glycan microarray technology. The data demonstrates that the tyrosine to asparagine substitution at position 137 does not impact NL03 binding, while lysine at position 145 is essential for binding as almost no binding was detected when this residue is Serine. The new data are reported in SI section 11, Fig. 13. These results demonstrate that the S145K mutation does not compensate the Y137N mutation but allows for a new hydrogen bond interaction that is detrimental for NL03 binding. We replaced the sentence by “*The MD simulation for NL91 HA (Supplementary Fig. 9b and 11) revealed that the S145K mutation at the sialic acid binding site enables a new hydrogen bond interaction with the OH4 of Neu5Ac. To probe the importance of this interaction, we created reverse mutations in NL03. Indeed, the K145 is essential for binding and when mutated to serine almost no response was detected when examined for binding on the glycan microarray (Supplementary Fig. 13). On the other hand, the Y137 side chain in NL91 HA is properly oriented to face the sialic acid-galactose*

glycosidic bond, which is consistent with the STD signals detected for H4 to H6s of this Gal residue (Fig. 3a) that were not observed for HK68 HA. We also confirmed the effect of this mutation by creating NL03 N137Y which did not significantly alter glycan binding on the array.”

Remark 4.2. To the best of my knowledge, Fernandez de Toro et al. first reported the use of STD NMR to study HA-receptor interaction in 2018 (PMID: 30238596). This study needs to be cited.

Response 4.3. We thank the reviewer for pointing out this paper. Although this paper does not deal with H3N2 viruses, we agree the methodology needs to be discussed. The following text has been included in the conclusion section. *“Recently, the binding of a symmetric sialylated-di-LacNAc containing N-glycan to an H1 protein was accomplished using an unnatural lanthanide tag to break the NMR signal degeneracy⁵². In this study, we addressed the problem of degeneracy by chemically synthesizing complex N-glycans having at specific positions ¹³C labeled monosaccharides without the need to introduce an artificial probe that may interfere in binding.”*

Remark 4.3. In the 2nd and 3rd paragraphs under section “Modelling of glycan-HA complexes”, many key HA-receptor interactions in recent H3N2 strains have identified by MD simulation. The authors need to compare these interactions identified by MD simulation with those previously observed in crystal structures, including PDB 6BKR (A/Wyoming/3/2003 + 6’-SLN, ref #32), PDB 6BKT (A/Michigan/15/2014 + 6’-SLN, ref #30), and PDB 6AOV (A/Brisbane/10/2007 + 6’-SLNLN, PMID: 29059230).

Response 4.3. We thank the Reviewer. We more compared in more detail the MD results with the structural reported data.

Minor comments:

1. In the first paragraph of introduction, it might be clearer to rephrase “12–56,000 deaths” as “12–56 thousand deaths”

The correction has been made.

2. "Indeed, the 2D 1H-STD-1H,13C-HSQC experiment with the three isotopomers 3, 4, and 5 showed substantially weaker 1H[13C]-STD signals than observed for NL03 HA (Fig. 5a)." It should be 5b instead of 5a.

We thank the Reviewer for notice the mistake. The corrected has been made.

3. "MD simulation revealed that N145 cannot engage the sialic acid by hydrogen bonding interaction in the way K145 does for NL03 HA (Supplementary Fig. 7d)." N145 is not labeled in Supplementary Fig. 7d.

We apologize for the mistake, the K145 is labelled in fFig. 7c, while the N145 is not in fig. 7d because it does not participate in binding. The correction has been made.

4. The authors should consider changing “H3Sing” to “Sing16”, so that the abbreviation scheme for influenza strain names is consistent throughout the manuscript (e.g. HK68, NL91, and NL03).

The correction has been made.

5. In the first paragraph of discussion, “recognition of sialosides by Has” should be “recognition of sialosides by HAs”.

The correction has been made.

REVIEWERS' COMMENTS

Reviewer #1 (Remarks to the Author):

All my concerns have been properly answered by the authors and to my opinion this is an excellent piece of work well deserving publication in Nature Communications.

Reviewer #3 (Remarks to the Author):

In the revised manuscript, the authors have sufficiently addressed my concerns in the initial review, as well as the other reviewers in my opinion. The revised manuscript clearly advances the field and the work will be of general interest to the scientific community.

Reviewer #4 (Remarks to the Author):

The authors did a great job in addressing my previous concerns. I only have one remaining concern. In Response 4.3, the authors claimed that Fernandez de Toro et al. (PMID: 30238596) did not deal with H3N2 viruses, and added the following text to the conclusion section: "Recently, the binding of a symmetric sialylated- di-LacNAc containing N-glycan to an H1 protein was accomplished using an unnatural lanthanide tag to break the NMR signal degeneracy⁵²". However, this statement is not true. Ref #52 (i.e. Fernandez de Toro et al., PMID: 30238596) did the NMR experiments using a recombinant H3 hemagglutinin from A/Hong Kong/1/1968 (HK/68) H3N2 influenza virus. FYI, here is the last sentence of the abstract of ref #52: "Furthermore, a detailed binding epitope of the tetradecasaccharide N-glycan in the presence of HK/68 hemagglutinin is described." The authors need to adjust their conclusion section accordingly.

Point-by-point response to the reviewers' comments:

Reviewer #1

All my concerns have been properly answered by the authors and to my opinion this is an excellent piece of work well deserving publication in Nature Communications.

Response: We thank this reviewer for this final comment.

Reviewer #3

In the revised manuscript, the authors have sufficiently addressed my concerns in the initial review, as well as the other reviewers in my opinion. The revised manuscript clearly advances the field and the work will be of general interest to the scientific community.

Response: We appreciate this reviewers comments.

Reviewer #4

The authors did a great job in addressing my previous concerns.

I only have one remaining concern. In Response 4.3, the authors claimed that Fernandez de Toro et al. (PMID: 30238596) did not deal with H3N2 viruses, and added the following text to the conclusion section: "Recently, the binding of a symmetric sialylated- di-LacNAc containing N-glycan to an H1 protein was accomplished using an unnatural lanthanide tag to break the NMR signal degeneracy⁵²". However, this statement is not true. Ref #52 (i.e. Fernandez de Toro et al., PMID: 30238596) did the NMR experiments using a recombinant H3 hemagglutinin from A/Hong Kong/1/1968 (HK/68) H3N2 influenza virus. FYI, here is the last sentence of the abstract of ref #52: "Furthermore, a detailed binding epitope of the tetradecasaccharide N-glycan in the presence of HK/68 hemagglutinin is described." The authors need to adjust their conclusion section accordingly.

Response: We are glad that we could address all earlier concerns.

The remaining issue has been corrected and it reads now as follows: ' Recently, the binding of a symmetric sialylated-di-LacNAc containing *N*-glycan to an H3 protein was accomplished using an unnatural lanthanide tag to break the NMR signal degeneracy⁵².'